# DHRL: A Graph-Based Approach for Long-Horizon and Sparse Hierarchical Reinforcement Learning

**Seungjae Lee**[1,2]**, Jigang Kim**[1,2]**, Inkyu Jang**[1,3]**, H. Jin Kim**[1,3]
[1]Seoul National University
[2]Artificial Intelligence Institute of Seoul National University (AIIS)
[3]Automation and Systems Research Institute (ASRI)
{ysz0301, jgkim2020, leplusbon, hjinkim}@snu.ac.kr

## Abstract

Hierarchical Reinforcement Learning (HRL) has made notable progress in complex control tasks by leveraging temporal abstraction. However, previous HRL algorithms often suffer from serious data inefficiency as environments get large. The extended components, $i.e.$, goal space and length of episodes, impose a burden on either one or both high-level and low-level policies since both levels share the total horizon of the episode. In this paper, we present a method of **D**ecoupling **H**orizons Using a Graph in Hierarchical **R**einforcement **L**earning (DHRL) which can alleviate this problem by decoupling the horizons of high-level and low-level policies and bridging the gap between the length of both horizons using a graph. DHRL provides a freely stretchable high-level action interval, which facilitates longer temporal abstraction and faster training in complex tasks. Our method outperforms state-of-the-art HRL algorithms in typical HRL environments. Moreover, DHRL achieves long and complex locomotion and manipulation tasks.

## 1 Introduction

Reinforcement Learning (RL) has been successfully applied to a range of robot systems, such as locomotion tasks [24, 8], learning to control aerial robots [10, 13], and robot manipulation [14, 22]. Goal-conditioned RL, which augments state with the goal to train an agent for various goals [23, 19], further raised the applicability of RL in robot systems allowing the agent to achieve diverse tasks.

Hierarchical Reinforcement Learning (HRL), which trains multiple levels of goal-conditioned RL, has improved the performance of RL in complex and sparse tasks with long horizons using temporally extended policy [25, 26, 16]. On the back of these strengths, HRL was adopted to solve various complex robotics tasks [20, 11, 18].

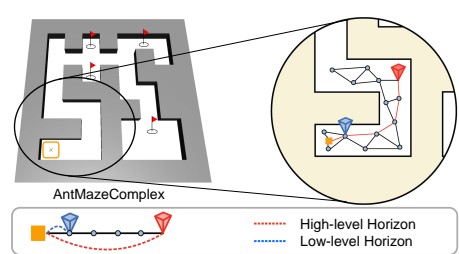

Figure 1: **DHRL:** By decoupling the horizons of both levels of the hierarchical network, DHRL not only solves long and sparse tasks but also significantly outperforms previous state-of-the-art algorithms.

However, HRL often has difficulty in complex or large environments because of training inefficiency. Previous studies speculated that the cause of this problem is the large goal space, and restricted the high-level action space to alleviate this phenomenon [28, 12]. Nevertheless, this approach performs well only in limited length and complexity and still suffers from the same trouble in larger environments.

36th Conference on Neural Information Processing Systems (NeurIPS 2022).

We show that this practical limitation of HRL can be mitigated by breaking down the coupled horizons of HRL. In previous HRL frameworks, the horizons of the low level and high level are related to each other structurally because they share the total length of the episode. This relation causes a tradeoff between the training burden of both levels; if the intervals between high-level actions increase (x-axis in Figure 2), the low-level policy has to cover a wider range, and in the opposite case, the high-level policy takes charge of the extended burden alone in large environments (y-axis in Figure 2). This is the reason why the previous HRL algorithms cannot cope with extended components of large environments (see Table 1 for the performance of the previous HRL method at various intervals).

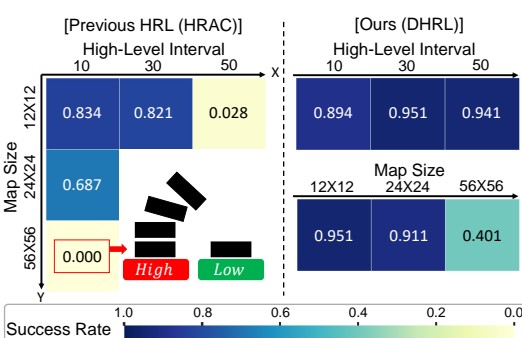

Figure 2: Our method is scalable in large environments by breaking down the relations between the two levels and allowing both levels to operate at their suitable horizons.

**We break down the coupled horizons of HRL:** To break the relation between the horizons of both levels, we adopt a graph structure. In our method, the high-level policy can use a longer temporal abstraction while the lower one only takes charge of smaller coverage by decomposing the subgoal into several waypoints with a graph. In this way, the HRL algorithm obtains the capability to stretch the interval of high-level action freely and achieves complex and large tasks, thanks to the enlarged strength of the HRL.

In summary, our main contributions are:

- We show that the previous HRL structures are not scalable in large environments, and that this limitation can be mitigated by removing coupled traits of high level and low level.

- To break down the coupled traits of HRL, we propose DHRL which decouples the horizons of high-level and low-level policies and bridges the gap using a graph.

- Our algorithm outperforms state-of-the-art algorithms in typical HRL environments and achieves complex and long tasks.

## 2  Preliminaries

We consider a finite-horizon Universal Markov Decision Process (UMDP) which can be represented as a tuple $(\mathcal{S}, \mathcal{G}, \mathcal{A}, \mathcal{T}, \mathcal{R}, \gamma)$ where $\mathcal{S}$, $\mathcal{A}$ and $\mathcal{G}$ are state space, goal space and action space respectively. The environment is defined by the transition distribution $\mathcal{T}(s_{t+1}|s_t, a_t)$ and reward function $\mathcal{R} : \mathcal{S} \times \mathcal{A} \times \mathcal{G} \rightarrow \mathbb{R}$, where $s_t \in \mathcal{S}$ and $a_t \in \mathcal{A}$ are the state and action at timestep $t$ respectively. Also, total return of a trajectory $\tau = (s_0, a_0, ..., s_H, a_H)$ is $R(\tau, g) = \sum_{t=0}^{H-1} \gamma^t r(s_{t+1}, g)$ where $r(s_{t+1}, g)$ (or $r(s_t, a_t, g)$) is a goal conditioned reward function and $\gamma$ is a discount factor. Subgoal $sg$, waypoint $wp$ and goal $g$ are defined in goal space $\mathcal{G}$ and we consider $\mathcal{G}$ that is a subspace of $\mathcal{S}$ with a mapping $\psi : \mathcal{S} \rightarrow \mathcal{G}$.

HRL framework typically has high-level policy $\pi^{\text{hi}}$ and low-level policy $\pi^{\text{lo}}$, each maintaining a separated replay buffer $\mathcal{B}^{\text{hi}}$ and $\mathcal{B}^{\text{lo}}$. Our method also follows the general HRL framework and employs both buffers that store high-level $(s_t, g_t, sg_t, r_t, s_{t+c_h}) \in \mathcal{B}^{\text{hi}}$ and low-level transition data $(s_t, wp_t, a_t, r_t, s_{t+1}) \in \mathcal{B}^{\text{lo}}$, where $c_h$ is the interval between high-level action.

One of the key problems in learning HRL is that the low-level policy $\pi^{\text{lo}}$ is non-stationary and thus old data from past policies may contain different next-states $s_{t+c_h}$ even though the high-level policy provides the same subgoal in the identical state. To bypass this off-policy discrepancy, HRL models use off-policy correction which relabels the old subgoal of high-level policy to be the most 'plausible subgoal' that will result in a similar transition in old data with the current low-level agent [17, 28, 12] (HIRO-style off-policy correction). Other approaches propose to relabel high-level action $sg_t$ to be achieved state $s_{t+c_h}$ (HAC-style hindsight action relabelling), reducing the computation cost to find a 'plausible subgoal' [15].

Graph-guided RL methods, which combine the strength of RL and planning by decomposing a long-horizon task into multi-step sub-problems, estimate the temporal distance between states and goals to construct a graph $\mathbf{G} = (\mathbf{V}, \mathbf{E})$ on goal space $\mathcal{G}$ without additional prior knowledge about environments. Previous studies proposed various methods to recover distance from Q-value [4, 9, 27]. If the agent gets -1 reward at every step except when it is in a goal area where the agent gets 0 reward, then, $Q_{\mathrm{lo}}(s, a|g)$ can reveal the temporal distance between $s$ to $g$ as: (Refer to the Appendix B for the detailed derivation.)

$$Dist(s \rightarrow g) = \log_\gamma \left( 1 + (1 - \gamma)Q_{\mathrm{lo}}(s, \pi(s, g)|g) \right) \tag{1}$$

However, it is known that recovering temporal distance correctly from the vanilla Q-network in this setting is challenging. For that reason, the previous methods use an additional Value function approximator [27] or distributional Q-networks [4].

## 3 Related work

**Graph-guided RL.** Graphs have recently been used as a non-parametric model in reinforcement learning (RL) to combine the advantages of RL and planning [4, 9, 27, 3, 6]. By decomposing a long-horizon task into multi-step planning problems, these studies have shown better performance and data efficiency. Search on the Replay Buffer (SORB) [4] constructs a directed graph based on the states randomly extracted from a replay buffer and Q-function-based edge cost estimation. The follow-up studies further improved the performance of earlier graph-guided RLs [4, 9] by combining additional methods such as graph search on latent space[27] or model predictive control [3].

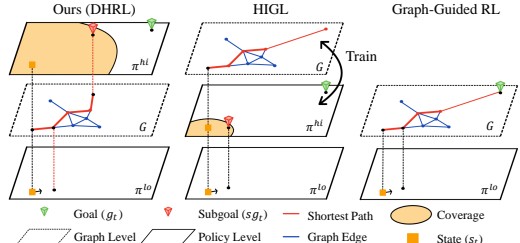

Figure 3: The differences between DHRL and the previous graph-based HRL methods. Our algorithm includes the graph structure between both levels explicitly while the previous methods use the graph only for training high-level policy or getting waypoints.

However, previous papers sidestepped the exploration problems in complex tasks through 'uniform initial state distribution' [4, 9, 27, 6], or work only on the dense reward settings [3]. We emphasize that the 'uniform initial state distribution' accesses privileged information about the environment during training by generating the agent uniformly within the feasible area of the map. This greatly reduces the scope of application of these algorithms. Unlike prior methods, ours can train from sparse reward settings and a 'fixed initial state distribution' without knowledge of the agent's surroundings, which makes it practical for physical settings. For detailed examples and comparison of various initial state distributions, see Table 2 and Figure 11 in Appendix C.

**Constrained-subgoal HRL.** To mitigate the training inefficiency issue of HRL, several researchers proposed methods that restrict the action of high-level agent to be placed in adjacent areas. Hierarchical Reinforcement Learning with k-step Adjacency Constraint (HRAC) [28] limits the subgoal to be in the adjacency space of the current state. Hierarchical reinforcement learning Guided by Landmarks (HIGL) [12] improved the data efficiency by adding novelty-based landmarks to adjacency-constrained HRL. However, these improvements only work on the limited length and complexity of the environment.

The previous work closest to our method is HIGL. However, there are three key differences between the previous work and our approach. First, our model explicitly includes the whole graph structure while HIGL needs to train high-level action to imitate the graph by using an additional loss term corresponding to the Euclidean distance from the nodes of the graph. Second, we use a graph to decouple the horizons of high level and low level, unlike the previous method which uses the graph only for guidance. Most importantly, ours can achieve goals in long and complex environments. To the best of our knowledge, there is no prior HRL research to train a model that has decoupled the horizons of the two levels.

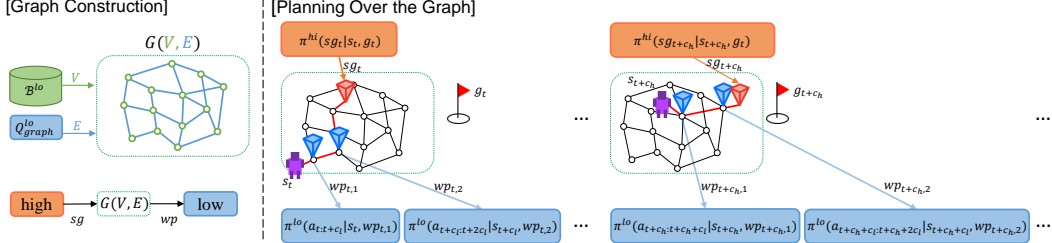

Figure 4: An overview of DHRL which includes the mid-level non-parametric policy between the high and low levels. The high-level (orange box) policy delivers subgoal $sg \in \mathcal{G}$ to the graph level (green box) and the graph instructs the low-level policy (blue box) to reach the waypoint $wp \in \mathcal{G}$. $c_h$ represents how long each high-level action operates for. The low level is given $c_{l,i}$ steps to achieve the goal where $c_h \neq c_{l,i}$.

# 4 Methods

We introduce Decoupling Horizons Using a Graph in Hierarchical Reinforcement Learning (DHRL), which can separate the time horizons of high-level and low-level policies and bridge the gap between both horizons using a graph. Our framework consists of high-level policy $\pi^{\mathrm{hi}}(sg|s,g)$, low-level policy $\pi^{\mathrm{lo}}(a|s,wp)$, and a graph $\mathbf{G}$. Given a goal $g_t$ in the environment, the high-level policy outputs a subgoal $sg_t$ (see Figure 4). Then, the shortest path from the current state $s_t$ to $sg_t$ is found on the graph. To do so, $s_t$ and $sg_t$ are added to the existing graph structure, then a sequence of waypoints $(s_t, wp_{t,1}, wp_{t,2}, ..., sg_t)$ is returned using a graph search algorithm. Finally, the low-level policy tries to achieve $wp_{t,i}$ during $c_{l,i}$ steps.

The key point of our method is that the low-level horizon $h^{\mathrm{low}} = c_l$ is unrelated to the high-level horizon $h^{\mathrm{high}}$. Since the high level generates one subgoal every $c_h$ steps, the H-step task is a $H/c_h$-step task for a high-level agent ($h^{\mathrm{high}} = H/c_h$) where $c_h$ is the interval between high-level action ($c_h > c_l$). In other words, unlike the previous HRL methods which have the relationship of

$$h^{\mathrm{high}} \times h^{\mathrm{low}} = H, \tag{2}$$

our algorithm does not have such relations, removing an obstacle toward a scalable-RL algorithm. Since the low-level horizon $c_l$ is determined by the edge cost between waypoints, we can also express the $c_l$ between the $i-1^{\mathrm{th}}$ waypoint and $i^{\mathrm{th}}$ waypoint as $c_{l,i}$, but we omit the letter $i$ in the later statements that do not need to specify the waypoint.

In section 4.1, we explain how to construct a graph over states and find a path on the graph level. In section 4.2, we present a low-level policy which can recover temporal distance between states without overestimation. In section 4.3, We introduce a strategy to train our method through graph-agnostic off-policy learning. Finally, in section 4.4, we propose additional techniques for better data efficiency in large environments.

## 4.1 Graph level: planning over the graph

This section details the graph search part in DHRL planning. We emphasize that every distance in the DHRL model is based on temporal distance $Dist(\cdot \rightarrow \cdot)$, obtained through Eq. (1). Thus, our algorithm requires no further information about the environment (*e.g.* Euclidean distance between states) than general HRL settings.

To find the shortest path, we adopt Dijkstra's Algorithm, as in the previous study [4]. The differences from the previous methods [4, 27] are the existence of the high-level policy, and whether the secondary path is considered. Let $\psi : \mathcal{S} \rightarrow \mathcal{G}$ be the projection of states on the goal space. In the graph initialization phase, we samples $n$ nodes (also called landmarks) using FPS algorithm [2] (Algorithm 2 in Appendix A) from $\psi(s)$ where $s \in \mathcal{S}$ is state sampled from $\mathcal{B}^{\mathrm{lo}}$. Then, we connect a directed edge $s_1 \rightarrow s_2$ if the temporal distance from $s_1$ to $s_2$ is less than the cutoff-threshold. In the planning phase, the graph $\mathbf{G}(\mathbf{V}, \mathbf{E})$ gets the subgoal $sg_t$ from the high-level policy and adds the projection of current state $\psi(s_t)$ and $sg_t$ to $\mathbf{V}$, so that the number of nodes in $\mathbf{G}$ becomes $n + 2$. Then, we connect the edges with costs less than the cutoff-threshold between the newly added nodes

and existing nodes. Next, we find the sequence of waypoints $\mathcal{W} : (wp_{t,0} = \psi(s_t), wp_{t,1}, wp_{t,2}, ..., wp_{t,k-1}, wp_{t,k} = sg_t)$ that connects from $\psi(s_t)$ to $sg_t$ using a graph search algorithm. At this time, if there is no path from $\psi(s_t)$ to $sg_t$, we adopt a secondary path from $\psi(s_t)$ to $sg_t^\dagger$, where $sg_t^\dagger$ is the closest node to $sg_t$ among the nodes connected from $\psi(s_t)$. After finding the waypoint sequence, the graph level provides $wp_{t,1}$ to the low-level policy and instructs it to reach $wp_{t,1}$. If it has been $Dist(wp_{t,i-1} \rightarrow wp_{t,i})$ steps since the agent started tracking $wp_{t,i}$ or it achieves $wp_{t,i}$, then we update the current tracking waypoint to the next waypoint $wp_{t,i+1}$. In order to reflect the newly discovered state, we update graph nodes at every $N$ episodes.

However, it is challenging to train Q-network which recovers reliable temporal distance. We explain the reason and a solution in the next section.

## 4.2 Low level: Separate Q-networks for Graph and Critic (SQGC)

Unlike previous graph-guided RL methods that leverage a graph only after the actor and critic networks have been learned, we construct and utilize a graph during training. To do so, the low-level policy $Q^{\text{lo}}$ is evaluated during training to assign edge costs. However, when $\pi^{\text{lo}}$ is not yet competent in achieving some goals or encounters difficult goals, the accumulation of failure experiences in the replay buffer causes underestimation of Q-values. This leads to an overestimation of temporal distance reconstructed from Eq. (1) and spoils the graph near the overestimated region by making the node-selection algorithm (Algorithm 2 in Appendix A) select more and more graph nodes around the overestimated area. Therefore, a temporal-distance reconstruction method is needed even when the policy is not sufficiently trained.

For this reason, we propose Separate Q-networks for Graph and Critic (SQGC) to prevent temporal distance overestimation. SQGC is composed of two identical Q-networks using different proportions of hindsight goal relabeling (HER) [1]. The SQGC includes $Q_{\text{critic}}^{\text{lo}}$ and $Q_{\text{graph}}^{\text{lo}}$ where $Q_{\text{critic}}^{\text{lo}}$ is for training $\pi^{\text{lo}}$, just like a typical application, and $Q_{\text{graph}}^{\text{lo}}$ is for recovering temporal distance between nodes. We substitute $wp_t$ with $\hat{wp}_t := ag_{t+t_{\text{ftr}}}$ in the sequential transition of a single episode $(s_t, wp_t, a_t, r(s_{t+1}, wp_t), s_{t+1})_{t=1:H-1}$ where $ag$ means the achieved goal and $t_{\text{ftr}}$ is a random integer drawn from the uniform distribution between 0 and $H - t$. To train $Q_{\text{graph}}^{\text{lo}}$, we relabel 100% of $wp_t$ in $(s_t, wp_t, a_t, r(s_{t+1}, wp_t), s_{t+1})_{t=1:H-1}$ as $\hat{wp}_t$, while we replace only 80% of $wp_t$ for $Q_{\text{critic}}^{\text{lo}}$.

Our method can prevent overestimation in the distance recovery by relabelling the goals in all transitions for training $Q_{\text{graph}}^{\text{lo}}$ because the experiences of failure are replaced with the successful trajectory (see section 5.3 for the ablation study). Also, by maintaining the original $Q_{\text{lo}}^{\text{critic}}$ to train $\pi_{\text{lo}}$, there is no degradation in the performance of the agent who might otherwise not be able to get negative feedback from failure since the failure will be relabeled as a desired goal.

However, it is still challenging to train HRL including a graph level using off-policy RL algorithms. We describe our approach to train DHRL using an off-policy algorithm in the next section.

## 4.3 High level: hindsight transitions for graph-agnostic off-policy learning

Thanks to decoupling the time horizons of both levels in HRL, the high-level policy in our method can look further without any additional burden on the low-level policy. In other words, We can stretch $c_h$, which represents how long each high-level action operates for. However, because of the extended interval and non-stationarity of the high-level MDP, it is challenging to train DHRL with an off-policy algorithm, which is important for data efficiency in that the off-policy algorithm can use the previous data from the replay buffer.

This non-stationarity of the high-level MDP is caused by the presence of a graph and low-level policy. Since the graph is gradually updated, it is challenging to train a model using an off-policy method from the data given by the previous graph. Furthermore, as $c_h$ gets longer, predicting the similarity of the trajectory of $c_h$ steps with only the first step of $\pi^{\text{lo}}(a_i|s_i, g_i)$ gets more difficult when we replace subgoal using HIRO-style off-policy correction [17] which is a popular method adopted by SOTA HRL algorithms [28, 12].

To facilitate the off-policy learning for longer $c_h$ and the changing graph in DHRL, we adopt a well-known hindsight action relabelling method proposed in Hierarchical Actor-Critic (HAC)

[15] in the high-level replay buffer data $(s_t, g_t, sg_t, r_t, s_{t+c_h}) \in \mathcal{B}^{\text{hi}}$. Alongside the original transitions data, we copy the transition data and replace the subgoal $(sg_t = \pi_\beta^{\text{high}}(sg_t|s_t, g_t))$ with the achieved goal $ag_{t+c_h}$ after $c_h$ steps. Thus, we use both transitions $(s_t, g_t, sg_t, r_t, s_{t+c_h})$ and $(s_t, g_t, ag_{t+c_h}, r_t, s_{t+c_h})$ to train the high-level policy. Our method utilizes the optimality of the graph and low-level policy to include a graph in the learning process, while the HAC-style hindsight action relabelling method assumes the optimal low-level policy $\pi^{\text{lo}*}$ only.

By replacing the previous subgoal with the achieved goal, we can assume that this transition was obtained from a stationary graph and $\pi^{\text{lo}*}$ with an error below a bound which is set to be a function of the density of the graph. We provide a theoretical analysis of the possibility of replacing the old off-policy graph with a virtual stationary graph in this section and Appendix B.

**Definition 4.1.** Given a compact state space $\mathcal{S}$, $\mathbf{G}(\mathbf{V}, \mathbf{E})$ is an $\epsilon-$resolution graph $if\ \forall s \in \mathcal{S}, \exists v \in V$ $s.t. \max(Dist(\psi(s) \rightarrow v), Dist(v \rightarrow \psi(s))) < \epsilon$, where $\mathbf{V} \subset \psi(\mathcal{S})$ and $E = \{(v_i, v_j)|v_i, v_j \in \mathbf{V}, Dist(v_i \rightarrow v_j) < c_l\}$.

**Theorem 4.2.** *Let $\mathbf{G}$ be an arbitrary $\epsilon-$resolution graph ($\epsilon < c_l/2$). Also, let $\pi_\beta^{\text{lo}}$ and $\mathbf{G}_\beta$ be the low-level policy and graph at the time of collecting the data. Off-policy error rate $\rho(\mathbf{G})$ is the normalized distance error with respect to the total traversal distance according to the change of $\pi_\beta^{\text{lo}}$ and $\mathbf{G}_\beta$ to $\pi^{\text{lo}*}$ and $\mathbf{G}$. If there is a path from $s$ to $g$, the upper bound of off-policy error rate $\rho(\mathbf{G})$ using a path obtained from graph search over $\mathbf{G}$ is $2\epsilon/c_l$.*

Theorem 4.2 shows that if we replace the previous subgoal with the achieved goal, and assume that these transitions are obtained from a stationary $\epsilon-$resolution graph $\mathbf{G}$, then the off-policy error $\rho$ is less than $2\epsilon/c_l$. Thus, we can train DHRL through a graph-agnostic off-policy RL algorithm using the substituted transition.

### 4.4 Optional techniques: gradual penalty and frontier-based goal-shifting

In this section, we propose two additional techniques, gradual penalty and frontier-based goal-shifting (FGS). These optional techniques can boost the performance of DHRL on some long tasks as shown in Figure 9(a) and (c). We note that these are not essential to train DHRL and ours outperforms previous HRL frameworks without these techniques (see section 5.3 for the ablation study).

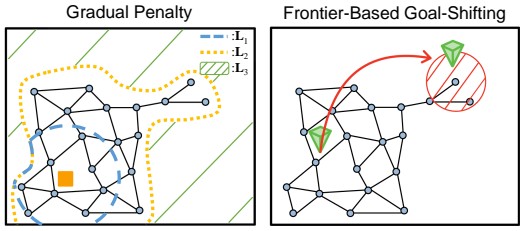

Figure 5: Optional techniques for better data efficiency: gradual penalty encourages the high-level policy to output a subgoal that the low-level agent can achieve. Frontier-based goal-shifting replaces goals in the explored area with new goals positioned in the rim of the graph.

**Gradual penalty.** Similar to the subgoal testing in the previous method [15], we propose gradual penalty (algorithm 4 in Appendix A), which can impose the penalty more delicately when the action space of the high-level agent gets large. We evaluate $sg_t$ of the original transition data $(s_t, g_t, sg_t, r_t, s_{t+c_h})$ by categorizing the following three cases; (a) close to the graph and low level actually achieved the subgoal $\in \mathbf{L}_1$, (b) close to the graph but the low-level policy could not achieve the subgoal $\in \mathbf{L}_2$ and (c) far from the graph $\in \mathbf{L}_3$. In this way, we can impose the more detailed penalties to respond to the expansion of the high-level action space as the high-level interval $c_h$ stretches.

**Frontier-based goal-shifting.** This optional technique has been devised to accelerate learning in a complex environment (algorithm 5 in Appendix A). FGS moves the final goal $g$ to the frontier area when $g$ comes into the place where the graph is already laid out during training. To check whether the goal is in the graph area, we examine whether $\min_{v \in \mathbf{V}}(Dist(v \rightarrow g))$ is smaller than cut-off threshold, where $\mathbf{V}$ is the set of the graph nodes. Alternative goals are the addition of random noise to nodes sampled from $v \in \mathbf{V}$ proportional to $-Q(s_0, \pi(s_0, v)|v)$. FGS is similar to the previous goal-directed exploration research [21] in that both use weighted samples, but ours does not maintain a generative model and samples the goals from the graph level in DHRL. Note that we do not use FGS when we compare DHRL with the previous state-of-the-art, since this FGS is beyond the main contribution, which is about HRL structure.

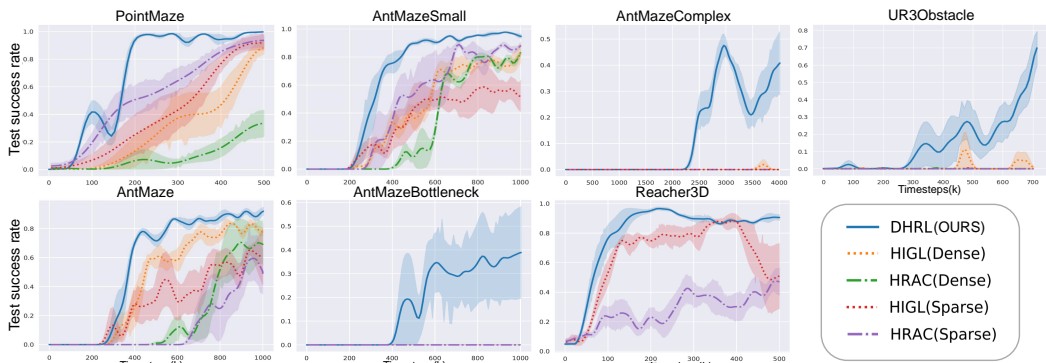

Figure 6: DHRL significantly outperforms prior state-of-the-art algorithms (success rate averaged over 4 random seeds and smoothed equally, and only the sparse settings for Reacher3D are tested as in the previous papers). Note that in AntMazeComplex, AntMazeBottleneck, and UR3Obstacle, the curves are not visible as they overlap at zero success rate.

## 5 Experiments

### 5.1 Environment description

We evaluate DHRL on robot environments based on the MuJoCo simulator including some sparse and long-horizon tasks. Firstly, various locomotion environments with 'fixed initial state distributions' are used to validate the temporal abstraction capability of DHRL in long-horizon and cluttered environments. In test episodes of the locomotion environments, the agent gets one of the most challenging goals (*i.e.*, the end of the maze). We also evaluate our algorithm in robot arm environments in which the agent aims to make the end-effector touch the goal, to evaluate our method in more complex dynamics.

- PointMaze / AntMazeSmall : The point / ant achieves the goal if it comes within 2.5 distance (success threshold) from the target point in $12 \times 12$ maze.
- AntMaze : $24 \times 24$ maze with success threshold 5
- AntMazeBottleneck : Bottleneck exists at the middle of the maze. The ant can barely pass through bottleneck.
- AntMazeComplex : $56 \times 56$ maze with success threshold 5
- Reacher3D : 7-dof robot arm aims to reach a goal.
- UR3Obstacle : 6-dof robot arm aims to reach a goal in an environment with several board-shaped obstacles.

We adopt TD3 algorithm [5] for high-level and low-level networks and use Dijkstra algorithm to find the shortest path in the graph level. Also, we note that only a few hyperparameters have been changed across various locomotion experiments: the number of nodes, penalty, and $c_h$. The results of DHRL in this paper are obtained using only sparse reward.

### 5.2 Experiment result

**Baselines.** We compare our method with state-of-the-art algorithms with and without a graph respectively; HIGL [12] and HRAC [28]. For more comparison with shallow RL (SAC) [7] and vanilla HRL (HIRO) [17], see Table 3 in Appendix C.

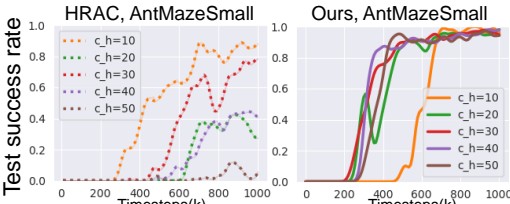

Figure 7: Robustness to $c_h$: HRAC vs Ours

**Comparison to state-of-the-art algorithms.**
Results are shown in Figure 6. Thanks to decoupling the horizons of the levels, DHRL could stretch the high-level interval and shows high data efficiency and success rate in various locomotion

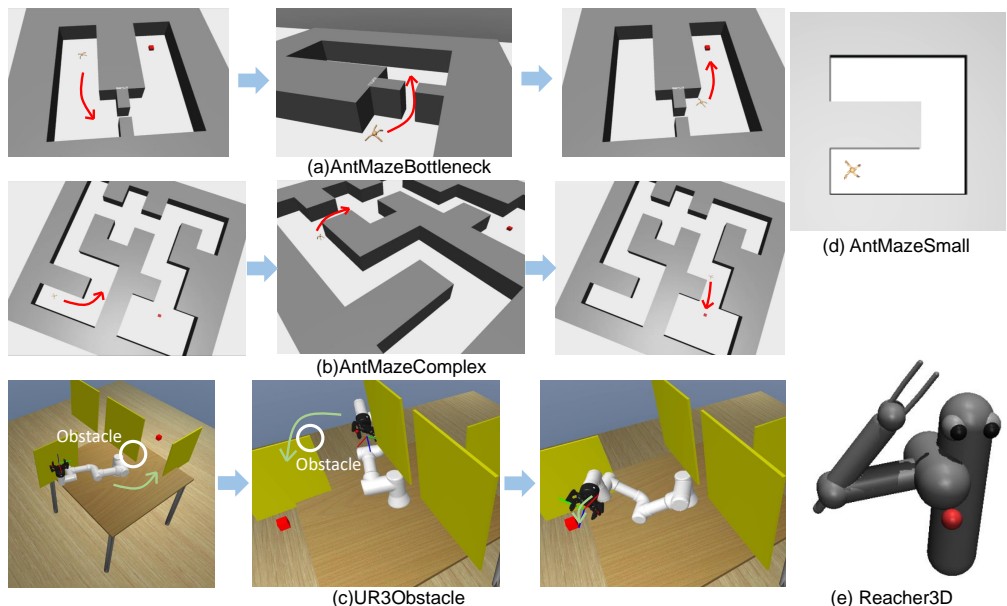

Figure 8: Locomotion and Reacher tasks in simulation: Additional to general tasks for HRL, we evaluated our method in long and sparse tasks (see (a), (b), and (c) in the figure). The maps of the environments are not given to the agent.

Table 1: The tradeoff in performance between the high level and the low level.

| Success Rate | HRAC $c_h = 5$ | HRAC $c_h = 10$ | HRAC $c_h = 30$ | HRAC $c_h = 50$ | **DHRL** |
|---|---|---|---|---|---|
| $12 \times 12$ map | 43.0% | **88.4%** | 78.3% | 4.5% | **95.1%** |
| $24 \times 24$ map | 18.0% | 48.9% | **57.4%** | 16.4% | **91.1%** |
| $56 \times 56$ map | 0.0% | 0.0% | 0.0% | 0.0% | **40.1%** |

and goal-reaching tasks. Moreover, DHRL is the only algorithm that can succeed in complex environments (AntMazeComplex and AntMazeBottleneck). When measuring the performance of the baselines, we selected $c_h$ with the best performance among the values $c_h$=10, 20, ..., and 80.

As shown in Figure 6, previous HRL methods cannot solve long-horizon tasks. This is likely due to the coupling ($c_h = c_l$) of the high-level horizon ($h^{\text{high}} = H/c_h$) and the low-level horizon ($h^{\text{low}} = c_l$) resulting in an increased burden on either the high-level or the low-level policy. For long-horizon tasks (large $H$), if $c_h(= c_l)$ is fixed, then $h^{\text{high}}$ increases and the high-level performance plunges (Table 1). On the other hand, if $c_h(= c_l)$ is increased, the low level has to manage a wider area and the performance plunges as shown in Figure 7 and Table 1.

### 5.3 Ablation study

**Decoupling horizons.** In this section, we examine how decoupling horizons affects the performance of long-horizon HRL and evaluate whether DHRL can stretch the interval of high-level policy. As shown in Figure 9(b), we tested various values of spacing of the high-level action, $c_h$. The result shows that even in long intervals of the high-level policy, DHRL shows consistent or improved performance (Figure 9(b)) without major degradation unlike previous HRL methods (Figure 7). In particular, DHRL agent in long-horizon environments such as AntMazeComplex shows that extended high-level interval is crucial for exploration and performance (see the rightmost figure in Figure 9(b)). Considering that the baselines could not solve long-horizon tasks even with various values of $c_h$, and the performance degrades with increasing $c_h$, we conclude that our method successfully decoupled the time steps in both levels and stretched the interval between high-level actions. This trait allows the high-level policy to look further and take advantage of the extended temporal abstraction.

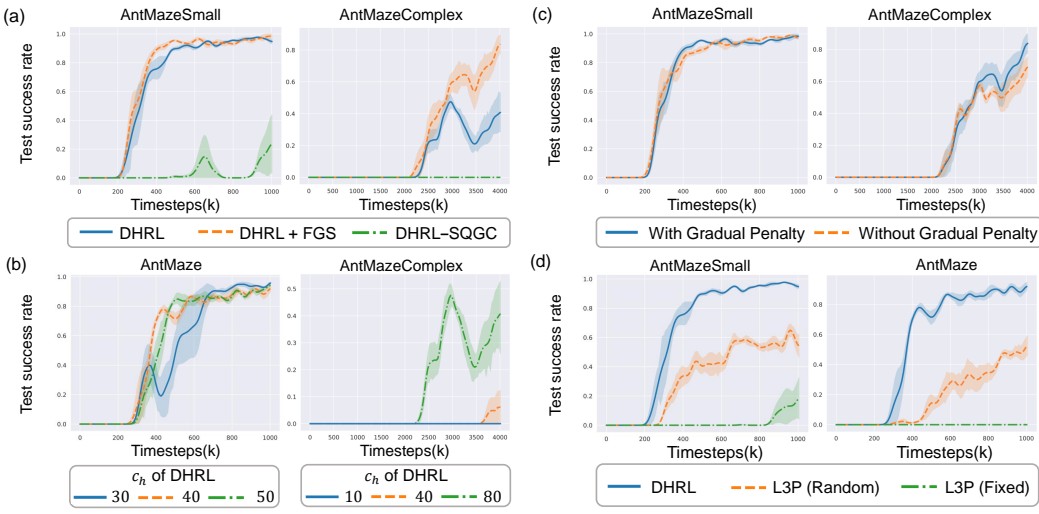

Figure 9: **Ablation study.** (a): The separate Q-network (SQGC) effectively alleviates the overestimation problem of temporal distance and FGS improves data efficiency. (b): Extending the high-level horizon is crucial in long-horizon tasks and DHRL successfully separates the horizons. (c): Gradual penalty improves the data efficiency in long tasks. (d): Without an additional high-level policy above the graph level, the performance plunges without the 'uniform initial state distribution'.

**Separate Q-network for Graph and Critic (SQGC).** In this section, we evaluate the effects of the Separate Q-network for Graph and Critic (SQGC) in our method. The red line in Figure 9(a), which is a variant of DHRL without an SQGC, clearly shows that the SQGC is critical to the performance. We empirically found that the overestimation of the temporal distance between states occurs especially near the obstacles (*e.g.* corners at the maze), and it is difficult for an agent to pass by without a separate Q network for graph construction. This is consistent with our analysis that the experiences of failure spoil the ability to recover the temporal distance from Q-network.

**Ablate high-level policy.** Graph-guided RL, which maintains graph level and low-level policy, is also a variant of HRL in that the graph level is a non-parametric version of the high-level policy (see Figure 3). From this point of view, the main difference between graph-guided RL and our method is the existence of the high-level policy. Then, why do we need additional high-level policy above the graph level?

Most graph-guided RL algorithms use 'uniform initial state distribution' to train the agent in complex environments and such assumption could be expensive, especially in the physical world where it is challenging to start from different positions each time. Figure 9(d) shows the result of the graph-guided RL algorithm, L3P [27], with a uniform and fixed initial state distributions. This indicates that without 'uniform initial state distribution', the performance and data efficiency of the previous graph-guided RL method drops seriously even in the smallest environment we experimented with. In contrast, DHRL can explore without 'uniform initial state distribution' even in long environments thanks to the enlarged temporal abstraction and exploration performance. Considering that the high-level policy suggests a subgoal to graph level, we conclude that the high level facilitates better exploration of the graph-guided RL. Thus, DHRL that has exploration capability also can be seen as the improved version of graph-guided RL.

**Optional techniques.** The result in Figure 9(c) shows that the gradual penalty can improve data efficiency in long tasks. By imposing a penalty on subgoals that are far from the current graph, the gradual penalty encourages the high-level agent to output a subgoal near the graph. Since the low-level agent can hardly achieve subgoals far from the explored area, the gradual penalty accelerates the training by providing more valid goals. FGS also can improve training efficiency in complex environments (Figure 9(a)). This means that it is important to provide goals near the unexplored areas which help the agent explore the unseen states in complex environments. With FGS, DHRL efficiently guides the agent by sampling the goals located on the rim of the graph.

# 6  Conclusion

We proposed a Decoupling Horizons Using a Graph in Hierarchical Reinforcement Learning (DHRL), which is a data-efficient HRL algorithm leveraging a graph to expand the range of problems that HRL can solve, by decoupling horizons and allowing both levels to operate at their suitable horizons. Our experimental results show that our method successfully separates the horizons of the levels and outperforms prior state-of-the-art methods. We believe that our method can present a direction towards scalable HRL allowing the hierarchical policy to take advantage of the extended temporal abstraction and have a lower dependency on the horizon of the environment. In this paper, we adopted a vanilla graph construction algorithm and sparse reward settings. We expect that if a novelty-based node selection is added to the graph construction, the performance could be further improved.

**Limitation.**  While our algorithm shows successful results in complex and long-horizon tasks, it might be difficult to construct a graph in some tasks that require complex interactions with the environment because of the higher-dimensional state space. We expect that this limitation can be solved by extending our algorithms into latent state space, or by incorporating a node sparsification algorithm, which we leave for future works.

# 7  Acknowledgement

This work was supported by AI based Flight Control Research Laboratory funded by Defense Acquisition Program Administration under Grant UD200045CD. Also, Seungjae Lee is grateful for financial support from Hyundai Motor Chung Mong-Koo Foundation.

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
