# A Algorithms.

---

**Algorithm 1** Training DHRL

---

1: sample $D_i^{\text{lo}} = (s_t, wp_t, a_t, r(s_{t+1}, wp_t), s_{t+1})_i \in \mathcal{B}^{\text{lo}}$
2: relabel $wp_t \leftarrow \hat{wp}_t = ag_{t+t_{\text{ftr}}}$ to make $\hat{D}_i^{\text{lo}}$
3: update $Q_{\text{critic},\theta_1}^{\text{lo}}$ and $\pi_{\phi_1}^{\text{lo}}$ using $D_i^{\text{lo}} \cup \hat{D}_i^{\text{lo}}$
4: update $Q_{\text{graph},\theta_2}^{\text{lo}}$ using $\hat{D}_i^{\text{lo}}$
5: **if** $t \bmod \tilde{d}$ **then**
6:      sample $D_i^{\text{hi}} = (s_t, g_t, sg_t, r_t, s_{t+c_h})_i \in \mathcal{B}^{\text{hi}}$
7:      relabel $sg_t \leftarrow ag_{t+c_h}$ to make $\hat{D}_i^{\text{hi}}$
8:      **for** $(s_t, g_t, sg_t, r_t, s_{t+c_h})$ in $D_i^{\text{hi}}$ **do**
9:        **if** $r(s_{t+c_h}, sg_t) = 0$ **then**
10:          $r_t \leftarrow r_t$      $\triangleleft$ subgoal $\in \mathbf{L}_1$
11:        **else if** use GradualPenalty **then**
12:          $r_t \leftarrow \texttt{GradualPenalty}(\text{graph } \mathbf{G}, sg_t, Q_{\text{graph}}^{\text{lo}})$
13:        **else**
14:          $r_t \leftarrow$ penalty $p_1$
15:        **end if**
16:      **end for**
17:      update $Q_{\theta_3}^{\text{hi}}$ and $\pi_{\phi_2}^{\text{hi}}$ using $D_i^{\text{hi}} \cup \hat{D}_i^{\text{hi}}$
18: **end if**

---

**Algorithm 2** Farthest Point Sampling Algorithm [2]

---

1: **Input:** set of states $\{s_1, s_2, ..s_K\}$, sampling number k, temporal distance function $Dist(\cdot \rightarrow \cdot)$
2: SelectedNode = []
3: DistList = [inf, inf, ... inf]
4: **for** $i = 1$ **to** $k$ **do**
5:     FarthestNode $\leftarrow$ argmax(DistList)
6:     add FarthestNode to SelectedNode
7:     DistFromFarthest $\leftarrow [Dist(\texttt{FarthestNode} \rightarrow s_1), ..., Dist(\texttt{FarthestNode} \rightarrow s_K)]$
8:     DistList = ElementwiseMin(DistFromFarthest, DistList)
9: **end for**
10: **return** SelectedNode

---

**Algorithm 3** Planning with DHRL

---

1: **while** not done **do**
2:     **if** t mod $graph\_construct\_freq$ **then**
3:        construct a graph $\mathbf{G}(\mathbf{V}, \mathbf{E})$ : sample $\mathbf{V} = \psi(s)$ where $s \in D^{\text{lo}}$ through FPS algorithm and get edge cost $\mathbf{E}$ by Eq. (1)
4:     **end if**
5:     $sg_t = \pi^{\text{hi}}(s_t, g_t)$
6:     get $\mathcal{W} : (wp_{t,0} = \psi(s_t), wp_{t,1}, wp_{t,2}, ..., wp_{t,k-1}, wp_{t,k} = sg_t)$
7:     previous waypoint index $idp \leftarrow 0$; tracking waypoint index $idt \leftarrow 1$; tracking time $t_{tr} \leftarrow 0$
8:     **for** $\tau = 1$ **to** $c_h$ **do**
9:        get low-level action $a_\tau$ from $\pi^{\text{lo}}(a_\tau | s_\tau, wp_{t,idt})$
10:        act $a_\tau$ in the environment and get $s_{\tau+1}$
11:        $t_{tr} += 1; t += 1$
12:        **if** agent achieve $wp_{t,idt}$ or $t^{tr} > Dist(wp_{t,idp} \rightarrow wp_{t,idt})$ **then**
13:          $idp += 1; idt += 1; t_{tr} \leftarrow 0$
14:        **end if**
15:     **end for**
16: **end while**

---

**Algorithm 4** Gradual Penalty

---

1: **Input:** graph $\mathbf{G}(\mathbf{V}, \mathbf{E})$, subgoal $sg_t$, $Q^{\text{lo}}_{\text{graph},\theta_2}$, gradual penalty threshold $\zeta_1$, penalty $p_1$, penalty $p_2$
2: **if** $\min(Q^{\text{lo}}_{\text{graph},\theta_2}(v \in \mathbf{V}, sg_t)) < \zeta_1$ **then**
3:    $r_t \leftarrow$ penalty $p_1$      $\triangleleft$ subgoal $\in \mathbf{L}_2$
4: **else**
5:    $r_t \leftarrow$ penalty $p_2$      $\triangleleft$ subgoal $\in \mathbf{L}_3$
6: **end if**

---

**Algorithm 5** Frontier-Based Goal-Shifting (FGS)

---

1: **Input:** $s_t$, graph $\mathbf{G}(\mathbf{V}, \mathbf{E})$, goal $g$, $Q^{\text{lo}}_{\text{graph},\theta_2}$, cut-off threshold $\zeta_2$
2: $Dist(s,g) := \log_\gamma \left(1 + (1-\gamma)Q^{\text{lo}}_{\text{graph},\theta_2}(s, \pi(s,g)|g)\right)$
3: **if** $\min_{v \in \mathbf{V}}(Dist(v \rightarrow g)) < \zeta_2$ **then**
4:    $\mathbf{V}_{\text{candidate}} \leftarrow \mathbf{V} + \text{noise}$
5:    $g_t \leftarrow \texttt{random.choice}(\mathbf{V}_{\text{candidate}}, \text{weight} = -Q^{\text{lo}}_{\text{graph},\theta_2}(s_t, \pi(s_t, \mathbf{V}_{\text{candidate}})|\mathbf{V}_{\text{candidate}}))$
6: **end if**
7: return $g_t$

---

**Algorithm 6** Overview of DHRL

---

1: **Input:** initial random steps $\tau_{\text{randomwalk}}$, initial steps without planning $\tau_{\text{w/o graph}}$, total training step $\tau_{\text{total}}$, Env, low-level agent $Q^{\text{lo}}_{\text{critic},\theta_1}$, $Q^{\text{lo}}_{\text{graph},\theta_2}$ and $\pi^{\text{lo}}_{\phi_1}$, high-level agent $Q^{\text{hi}}_{\theta_3}$ and $\pi^{\text{hi}}_{\phi_2}$
2: $Dist(s,g) := \log_\gamma \left(1 + (1-\gamma)Q^{\text{lo}}_{\text{graph},\theta_2}(s, \pi(s,g)|g)\right)$
3: **for** $\tau = 1$ **to** $\tau_{\text{total}}$ **do**
4:    **if** Env.done **then**
5:       Env.reset (episode step resets to 0)
6:       **if** Use FGS **then**
7:          $g \leftarrow \text{FGS}(\mathbf{G}, g, Q^{\text{lo}}_{\text{graph},\theta_2})$
8:       **end if**
9:    **end if**
10:   **if** $\tau < \tau_{\text{randomwalk}}$ **then**
11:      $a_t \leftarrow$ random.uniform(high = action.high, low = action.low)    $\triangleleft$ random action
12:   **else if** $\tau < \tau_{\text{w/o graph}}$ **then**
13:      $a_t \leftarrow$ vanilla $HRL(sg_t = \pi^{\text{hi}}_{\phi_2}(s_t, g)$ and $\pi^{\text{lo}}_{\phi_1}(s_t, sg_t))$    $\triangleleft$ act without planning
14:   **else**
15:      **if** Graph $\mathbf{G}$ is not initialized **then**
16:         Create a graph $\mathbf{G}(\mathbf{V}, \mathbf{E})$ using FPS algorithm    $\triangleleft$ initialize graph
17:      **end if**
18:      **if** episode step(the step of the environment) $\% c_l = 0$ **then**
19:         $sg_t \leftarrow \pi^{\text{hi}}_{\phi_2}(s_t, g)$    $\triangleleft$ get subgoal
20:         $\{wp_{t,1}, wp_{t,2}, \cdots wp_{t,k}\} \leftarrow \texttt{Dijkstra'salgorithm}(s_t, sg_t)$   $\triangleleft$ get waypoints
21:         current waypoint index $n = 1$
22:      **end if**
23:      **if** achieved $wp_{t,n}$ or tried more than $Dist(wp_{t,n-1}, wp_{t,n})$ to achieve $wp_{t,n}$ **then**
24:         current waypoint index $+= 1$
25:      **end if**
26:      $a_t \leftarrow \pi^{\text{lo}}_{\phi_1}(s_t, wp_{t,n+1})$    $\triangleleft$ get low-level action
27:   **end if**
28:   Env.step($a_t$)
29:   Train low-level agent $Q^{\text{lo}}_{\text{critic},\theta_1}$, $Q^{\text{lo}}_{\text{graph},\theta_2}$ and $\pi^{\text{lo}}_{\phi_1}$, high-level agent $Q^{\text{hi}}_{\theta_3}$ and $\pi^{\text{hi}}_{\phi_2}$
30:   **if** $\tau \%$ graph update freq $= 0$ **then**
31:      Update Graph $\mathbf{G}(\mathbf{V}, \mathbf{E})$ using FPS algorithm
32:   **end if**
33: **end for**

---

# B  Proofs of Theorems.

**Derivation of equation 1.**  If a given policy $\pi_{\text{lo}}$ requires n steps to get from current $s$ to a goal $g$, the $\gamma$-discounted return is $Q_{\text{lo}}(s, \pi(s, g)|g) = (-1) + (-1)\gamma + (-1)\gamma^2 \cdots (-1)\gamma^{n-1} = -\frac{1-\gamma^n}{1-\gamma}$.

Thus, the temporal distance between $s$ to $g(= n)$ is derived from $\gamma^n - 1 = (1 - \gamma)Q_{\text{lo}}(s, \pi(s, g)|g)$ as

$$n = \log_\gamma \left(1 + (1 - \gamma)Q_{\text{lo}}(s, \pi(s, g)|g)\right). \tag{3}$$

**Definition B.1.**  $\mathcal{W}_{\mathbf{G}}(s_t, sg_t) = (wp_{t,0}, wp_{t,1}, ..., wp_{t,k})$ is a sequence of waypoint obtained by the graph search algorithm and $w(\mathcal{W}_{\mathbf{G}}, \tau) = wp_{t,i} \in \mathcal{W}_{\mathbf{G}}(s_t, sg_t)$ is the waypoint that is given to low-level policy at $\tau$.

Given the transition distribution of the environment $\mathcal{T}(s_{\tau+1}|s_\tau, a_\tau)$, the transition data $(s_t, g_t, sg_t, r(s_{t+c_h}, g_t), s_{t+c_h})$ from the high-level policy's replay buffer has been obtained as

$$s_{t+c_h} = \prod_{\tau=t}^{t+c_h-1} \mathcal{T}(s_{\tau+1}|s_\tau, a_\tau) \cdot \pi_\beta^{\text{lo}}(a_\tau|s_\tau, w(\mathcal{W}_{\mathbf{G}_\beta}, \tau)), \tag{4}$$

where $\pi_\beta^{\text{lo}}$ and $\mathbf{G}_\beta$ are the previous low-level policy and graph respectively. Also, by using a different graph $\mathbf{G}$ and an optimal policy $\pi^{\text{lo}*}$, we get a new transition data $(s_t, g_t, sg_t, r(s'_{t+c_h}, g_t), s'_{t+c_h})$, where

$$s'_{t+c_h} = \prod_{\tau=t}^{t+c_h-1} \mathcal{T}(s_{\tau+1}|s_\tau, a_\tau) \cdot \pi^{\text{lo}*}(a_\tau|s_\tau, w(\mathcal{W}_{\mathbf{G}}, \tau)). \tag{5}$$

For given $s_t$ and $s_{t+c_h}$, we define the off-policy error rate, which is the normalized distance error with respect to the total traversal distance according to the change of $\pi_\beta^{\text{lo}}$ and $\mathbf{G}_\beta$ to $\pi^{\text{lo}*}$ and $\mathbf{G}$, as

$$\rho(\mathbf{G}) = \frac{Dist(\psi(s'_{t+c_h}) \to \psi(s_{t+c_h}))}{Dist(\psi(s_t) \to \psi(s_{t+c_h}))}. \tag{6}$$

**Lemma B.2.**  *Suppose that $Dist(\cdot \to \cdot)$ in Eq. (1) is Lipschitz continuous. Then, there exists a constant $L > 0$ such that $\forall x$ and $y$, $\max(Dist(x \to y), Dist(y \to x)) \leq L||x - y||$, where $|| \cdot ||$ is the Euclidean norm, since $Dist(x \to x) = 0$. Then, any $\epsilon/L-$resolution graph w.r.t the Euclidean norm, whose existence is trivial, is an $\epsilon-$resolution graph w.r.t $Dist(\cdot \to \cdot)$.*

**Proof of Theorem 4.2**

*Proof.* Let $\mathcal{C}^{s \to g}$ be one of the shortest paths from $s$ to $g$ and $T$ be the distance of $\mathcal{C}^{s \to g}$. Also let $p_1 \in \mathcal{C}^{s \to g}$ be a point that $Dist(\psi(s) \to p_1) = c_l - \epsilon$. Then, $\exists wp_1 \in \mathbf{V}$ s.t. $\max(Dist(p_1 \to wp_1), Dist(wp_1 \to p_1)) < \epsilon$, because $\mathbf{G}$ is an $\epsilon-$resolution graph. Since $Dist(\cdot \to \cdot)$ is a temporal distance, it satisfies the triangular inequality and then, $Dist(\psi(s) \to wp_1) \leq Dist(\psi(s) \to p_1) + Dist(p_1 \to wp_1) < (c_l - \epsilon) + \epsilon = c_l$ and $Dist(wp_1 \to g) \leq Dist(wp_1 \to p_1) + Dist(p_1 \to g) < \epsilon + (T - c_l + \epsilon) = T - c_l + 2\epsilon$.

Repeating the above procedure, let $p_{i+1} \in \mathcal{C}^{wp_i \to g}$ be a point that $Dist(wp_i \to p_{i+1}) = c_l - \epsilon$. Then, $\exists wp_{i+1} \in \mathbf{V}$ s.t. $\max(Dist(p_{i+1} \to wp_{i+1}), Dist(wp_{i+1} \to p_{i+1})) < \epsilon$. Then, $Dist(wp_i \to wp_{i+1}) < c_l$ and $Dist(wp_{i+1} \to g) < T - (i + 1)c_l + 2(i + 1)\epsilon$. Consequently, the agent after $T$ time-step will be closer than the $\lfloor T/c_l \rfloor^{th}$ waypoint from $g$. The remaining distance is less than

$$T - \lfloor T/c_l \rfloor c_l + 2\lfloor T/c_l \rfloor \epsilon. \tag{7}$$

Thus, if an agent follows the sequence of waypoints $\{s, wp_1, wp_2, ..., g\}$, which is generated from a graph search algorithm over $\mathbf{G}$ and $\pi^{\text{lo}*}$, the error rate over this path satisfies

$$\rho(\mathbf{G}) \leq \frac{T - \lfloor T/c_l \rfloor c_l + 2\lfloor T/c_l \rfloor \epsilon}{T} \leq \frac{T - (c_l - 2\epsilon)(T/c_l)}{T} = \frac{2\epsilon}{c_l}. \tag{8}$$

Thus the off-policy error rate $\rho$ is equal or less than $2\epsilon/c_l$ during $T$. Since all path from $s$ to $g$ takes at least $T$ time-steps, this upper-bound of error rate is also satisfied in all path from $s$ to $g$. $\square$

# C Additional Results

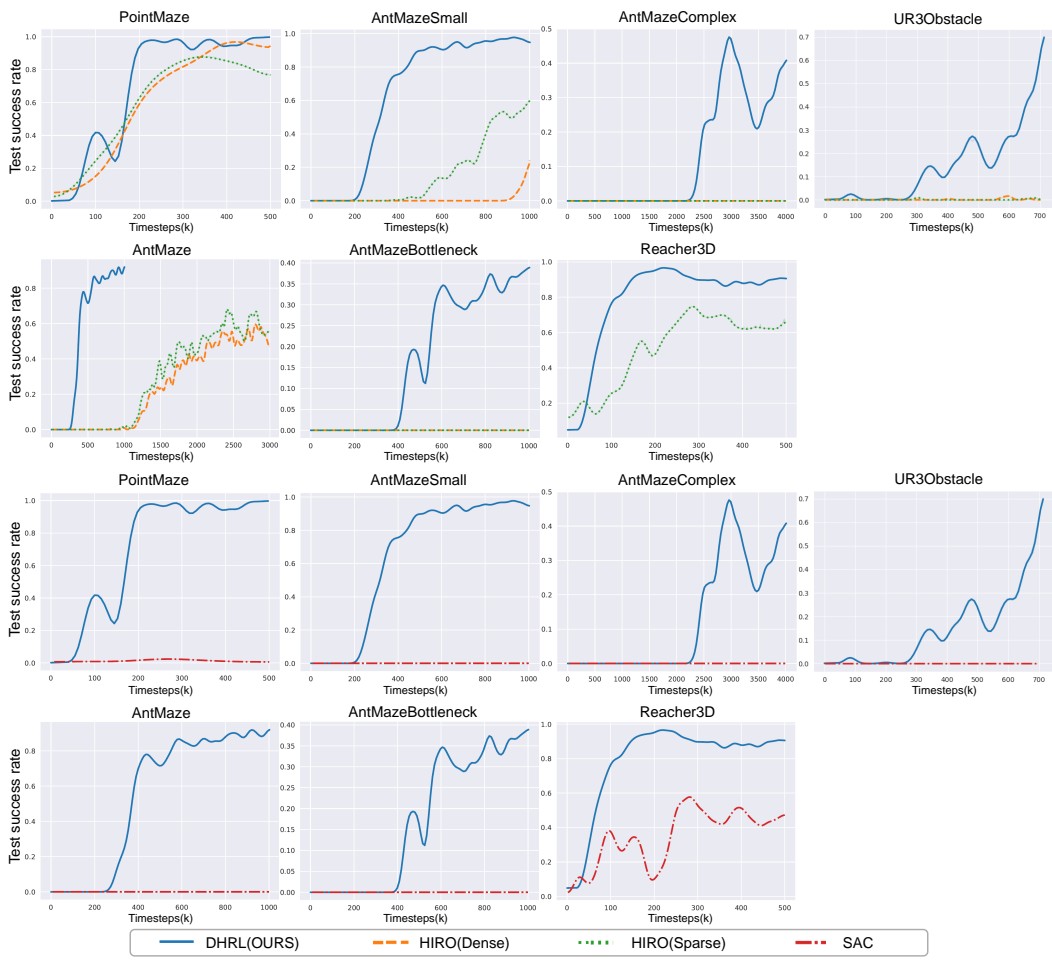

Figure 10: Comparison with shallow RL (SAC) and vanilla HRL (HIRO). The completely failed baselines are occluded by others.

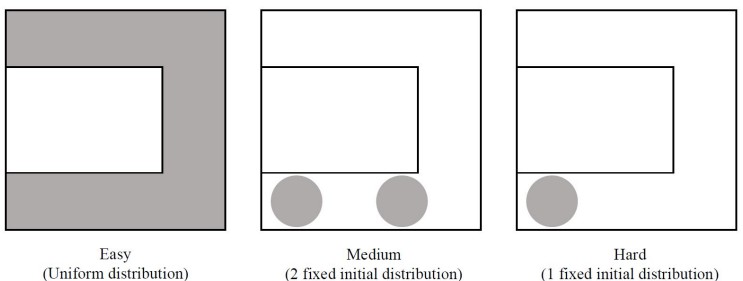

Easy
(Uniform distribution)

Medium
(2 fixed initial distribution)

Hard
(1 fixed initial distribution)

Figure 11: Examples of various initial state distributions.

Table 2: Performance of DHRL in various difficulties of initial state distributions.

| SUCCESS RATE | EASY(UNIFORM) | MEDIUM(2 FIXED POINT) | HARD(1 FIXED POINT) |
|---|---|---|---|
| ANTMAZE 0.3M | 80.4% | 28.5% | 12.2% |
| ANTMAZE 0.5M | 87.1% | 88.2% | 71.5% |

As shown in the table above, the wider the initial distribution, the easier it is for the agent to explore the map. In other words, the 'fixed initial state distribution' condition we experimented with in this paper is a more difficult condition than the 'uniform initial state distribution' that previous graph-guided RL algorithms utilize. Of course, 'fixed initial state distribution' requires less prior information about the state space. We further experimented with ours (DHRL) under various types of reset conditions as shown in Table 2. As expected, our algorithm shows faster exploration at the uniform reset point.

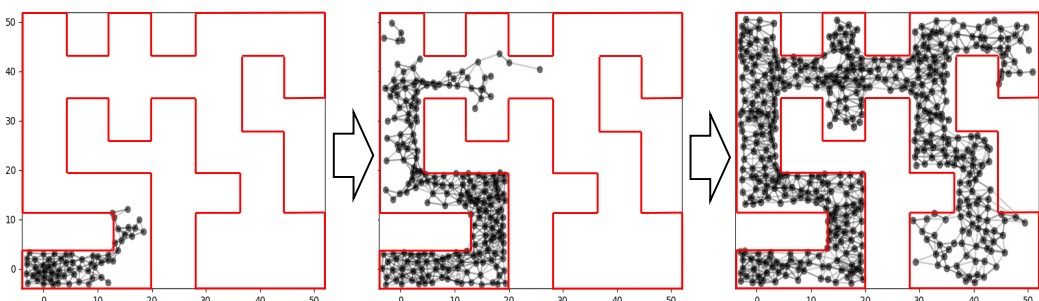

Figure 12: Changes in the graph level over the training; DHRL can explore long tasks with 'fixed initial state distribution' and limited knowledge about the environment.

Table 3: **Comparisons between our algorithm (DHRL) and baselines:** The numbers next to the environment names are the time-steps for training the models. The results are averaged over 4 random seeds and smoothed equally. '-D' and '-S' mean dense reward and sparse reward respectively. We use NVIDIA RTX A5000.

| SUCCESS RATE | | SAC | HIRO-D | HIRO-S | HRAC-D | HRAC-S | HIGL-D | HIGL-S | DHRL |
|---|---|---|---|---|---|---|---|---|---|
| POINTMAZE | 0.25M | 1.2% | 73.8% | 77.9% | 6.1% | 56.1% | 25.5% | 34.3% | **96.9%** |
| | 0.5M | 0.3% | 93.6% | 76.8% | 33.5% | 93.7% | 88.3% | 91.8% | **99.8%** |
| ANTMAZESMALL | 0.5M | 0.0% | 0.0% | 2.0% | 9.2% | 54.0% | 36.3% | 44.1% | **89.8%** |
| | 1.0M | 0.0% | 24.1% | 60.1% | 83.4% | 88.4% | 83.7% | 52.2% | **95.1%** |
| ANTMAZE | 0.5M | 0.0% | 0.0% | 0.0% | 1.1% | 0.0% | 60.2% | 32.7% | **71.5%** |
| | 1.0M | 0.0% | 0.7% | 0.8% | 68.7% | 48.9% | 78.1% | 60.3% | **91.1%** |
| BOTTLENECK | 0.5M | 0.0% | 0.0% | 0.0% | 0.0% | 0.0% | 0.0% | 0.0% | **16.5%** |
| | 1.0M | 0.0% | 0.0% | 0.0% | 0.0% | 0.0% | 0.0% | 0.0% | **38.7%** |
| COMPLEX | 2.5M | 0.0% | 0.0% | 0.0% | 0.0% | 0.0% | 0.0% | 0.0% | **20.4%** |
| | 4.0M | 0.0% | 0.0% | 0.0% | 0.0% | 0.0% | 0.0% | 0.0% | **40.1%** |
| REACHER-3D | 0.25M | 49.2% | - | 66.0% | - | 26.5% | - | 78.2% | **95.1%** |
| | 0.5M | 47.2% | - | 67.1% | - | 44.1% | - | 47.1% | **90.6%** |
| UR3OBSTACLE | | 0.0% | 2.5% | 1.9% | 0.5% | 0.0% | 11.1% | 0.5% | **69.8%** |

Table 4: Hyperparameters for HRL: When evaluating the previous HRL algorithms, we used the same hyperparameters as used in their papers. We also tried various numbers of landmarks and $c_h$ which may affect the performance in long-horizon tasks.

|  | HIRO | HRAC | HIGL |
|---|---|---|---|
| high-level $\tau$ | 0.005 | 0.005 | 0.005 |
| $\pi^{\mathrm{hi}}$ lr | 0.0001 | 0.0001 | 0.0001 |
| $Q^{\mathrm{hi}}$ lr | 0.001 | 0.001 | 0.001 |
| high-level $\gamma$ | 0.99 | 0.99 | 0.99 |
| high-level train freq | 10 | 10 | 10 |
| $c_h$ |  | 10-50 |  |
| low-level $\tau$ | 0.005 | 0.005 | 0.005 |
| $\pi^{\mathrm{lo}}$ lr | 0.0001 | 0.0001 | 0.0001 |
| $Q^{\mathrm{lo}}$ lr | 0.001 | 0.001 | 0.001 |
| low-level $\gamma$ | 0.95 | 0.95 | 0.95 |
| hidden layer | (300,300) | (300,300) | (300,300) |
| number of coverage landmarks $\gamma$ | - | - | 20-250 |
| number of novelty landmarks $\gamma$ | - | - | 20-250 |
| batch size | 128 | 128 | 128 |

Table 5: Hyperparameters for SAC

|  | SAC |
|---|---|
| hidden layer | (256, 256, 256) |
| actor lr | 0.0003 |
| critic lr | 0.0003 |
| entropy coef | 0.2 |
| $\tau$ | 0.005 |
| batch size | 256 |
| $\gamma$ | 0.99 |

Table 6: Hyperparameters for DHRL

|  | DHRL |
|---|---|
| hidden layer | (256, 256, 256) |
| initial episodes without graph planning | 75 |
| gradual penalty transition rate | 0.2 |
| high-level train freq | 10 |
| actor lr | 0.0001 |
| critic lr | 0.001 |
| $\tau$ | 0.005 |
| $\gamma$ | 0.99 |
| number of landmarks | 300-500 |
| target update freq | 10 |
| actor update freq | 2 |