# OpenReview forum: "DHRL: A Graph-Based Approach for Long-Horizon and Sparse Hierarchical Reinforcement Learning"
_NeurIPS.cc/2022/Conference — NeurIPS 2022 Accept_

### Official Review · Reviewer_ctmE · 2022-07-11

**Rating:** 7
**Confidence:** 4
**Soundness:** 4 excellent
**Presentation:** 3 good
**Contribution:** 3 good

**Summary:**

The paper proposes to use a graph construction and search method in Hierarchical RL to break down subgoals from the high-level policy into fine-grained subgoals for the low-level policy. Essentially, the paper proposes to use 3 levels: the highest and lowest levels use RL methods and the middle level is using a graph, which is built using learned temporal distances.
In this way, the horizons of the levels are more decoupled and long-horizon problems can be solved. Its benefits are shown in several simulated robotic environments such as maze tasks and manipulator reaching tasks.

**Questions:**

- Can you report the numbers of just the graph+lowlevel (so essentially L3P) AFTER you learned the graph with your method. I expect the numbers to be as high as with your method, which would be valuable information for further research.

- Please clarify Eq 1.

- What is the size of the graphs that are constructed? (is it the number of landmarks?)

- There are threshold parameters for reaching a waypoint/subgoal by the low-level. I did not see the parameter used or this. Also how sensitive is the method to this choice?

**Ethics Review Area:**

["I don’t know"]

**Limitations:**

Limitations have been addressed

**Strengths And Weaknesses:**

Strengths:
- the method is sound
- the performance is much better than the baselines
- ablations were conducted on most of the components


Weaknesses:
- the language of the paper needs improvements
- the clarify can be improved
- I suspect, the high level is only relevant for exploration and is potentially not required for solving the task.

Details:
- You are missing a lot of articles in your text, so many occurrences, that I will not mention them here. For instance: ... Using Graph in HRL -> ... Using a Graph
- you should replace: "extremely long" with "very long" or just "long"
- L28: using a temporally...
- L80: I think it is not a great practice to use two-letter variable names in mathematics (sg)
- Eq 1: I am wondering whether there is maybe a sign error in EQ 1. Maybe, I am mistaken, but I thought it should be $\log_\gamma (1 - (\gamma-1)Q)$. I think the derivation of that formula might be nice in the Appendix.
- L94: Graph as recently.... ->  Graphs have recently been used ...
- L106ff: citations missing
- Sec 3: I think you should compare your method to multilevel hierarchical methods as you have 3 levels
   A method that also has not a fixed time horizon for the low level is HiTS [1] but it does not solve the long-horizon problem.
- L139: why is c_l also depending on "i"? Later you only have c_l without i. c_h is also not defined unless I have overlooked it.
- L183: We -> we
- L188: formulation with "whole" and "part" is not ideal. You mean all occurrences vs a subset (here 80%)...
- L213: how is that different from HAC? You write it is inspired, but to me, it looks just like HAC.
- L220: why are "forall" and "exists" in superscript?
- L231: I thought the problem is not off-policy correction, but a non-stationarity of the semi-MDP.
- L241ff: I am missing the details here. I could not implement it from what is written here. Also for FGS, more details should be provided (in Appendix and references, for instance).
- L268: A threshold of 2.5 in a 12x12 maze is really a huge goal area. Is that standard? Similar in the AntMaze. What is the size of the Ant? A threshold of 5 is really big. Please specify what is a standard-settings and what you changed. If you can solve it with a smaller threshold, that might also be a strong additional experiment.

L299: horizon^{high} maybe h^{high}. Also, such words should not be italic. What is the difference between c and horizon?
L360: grammar

[1] Hierarchical Reinforcement Learning with Timed Subgoals, Gürtler et. al, NeurIPS 2021

-- post rebuttal --
most of my concerns are addressed and I am raising my score.

---

> ### Author Response · Authors · 2022-08-02
> **Response to reviewer ctmE (1/2)**
>
> Dear reviewer ctmE,
>
>
> We sincerely appreciate your constructive and insightful comments. We found them extremely helpful in improving our manuscript. We prepared our response below:
> - - -
> **Q1. What is a standard-settings of success threshold? Can DHRL solve it with a smaller threshold?**
>
> We note that two SOTA baselines used exactly the same success threshold as ours.(HRAC,HIGL). Also, following your suggestion, we additionally evaluated our algorithm over various success thresholds. Interestingly, DHRL maintains superior performance without significant performance degradation even on the reduced (i.e. more strict) success criteria. As you mentioned, we believe that this result also indicates a strong merit of our algorithm.
>
> | success threshold | 0.5 | 2.5 | success threshold | 1.0 | 5.0 |
> | --- | --- | --- | --- | --- | --- |
> | AntMazeSmall | 0.936 | 0.948 | AntMaze | 0.900 | 0.912 |
> - - -
> **Q2. Derivation of the formula 1**
>
> **A2.** Thank you for correcting the typo in the equation 1. Following your suggestion, we added the derivation of the formula to the appendix.
> - - -
> **Q3. What is the size of the graphs that are constructed? Is it the number of landmarks?**
>
> **A3.** Yes, the size of graphs is the number of landmarks (or nodes).  These nodes are selected through the FPS algorithm (Algorithm 2 in Appendix A).
> - - -
> **Q4. Evaluation without high-level policy AFTER the graph is learned**
>
> **A4.** Following your suggestion, we evaluated our algorithm without using the high-level policy AFTER the graph is learned. To do so, we chose networks with well-trained graphs for each environment and evaluated them with and without high-level policy. As shown in the table below, DHRLs with high-level agents have a higher success rate than those without graphs (about 9% of improvement).
>
> | success rate | antmaze | bottleneck |
> | --- | --- | --- |
> | train with high-level policy, do NOT remove high-level policy | **0.973** | **0.946** |
> | train with high-level policy, remove it AFTER graph learned | 0.892 | 0.865 |
> | L3P | 0.556 | 0.000 |
>
> Given that exploration is one of the very important components in RL, and that our project is motivated to maximize this exploration performance by allowing the high-level policy to look further, the high-level agent in DHRL is not any less significant than in other HRL. It is already well known that the benefit of hierarchy is due to exploration[1]. Also, [1] has shown that the non-hierarchical agent trained on experience collected by an HRL agent can achieve performance competitive with HRL. **As this fact does not serve as a 'weakness' of HRL research, the fact that '(DHRL without high level AFTER graph is trained) shows good performance' could not be a weakness of DHRL.**
>
> Also, it should be noted that without a high-level policy, training well-formed graphs are very challenging. To train such a well-formed graph without high-level policy, the RL agent requires additional modules to increase the exploration ability in a way such as a goal generation algorithm or curriculum method (such as Skew-fit or HGG), which means that training an additional neural network is necessary.
>
> We expect that these findings will make a significant improvement in the applicability of our algorithm since it provides one more option: to fully use the performance of DHRL, the high-level policy can be used, and for lighter models, DHRL-without_high option can be used.
> - - -
> **Q5. clarification of terms : $c_h$ and $c_l$**
>
> Following your suggestion, we changed the notation of high-level horizon ( $horizon^{high}$) and low-level ( $horizon^{low}$) to $h^{high}$ and $h^{low}$.
>
> **A5-1.** $c_h$ is the interval between high-level action, and $c_l$ is the low-level horizon. In previous HRL algorithms, $c_h=c_l$. Since the high level generates one subgoal every $c_h$ steps, H-step task is a ${H}/{c_h}$-step task for the high-level agent (i.e. $h^{high} = {H}/{c_h}$). Also, of course, $h^{low} = c_l$.
>
> *high-level action: a subgoal the graph level receives
>
> *low-level horizon: the number of steps a low-level agent spends trying to achieve a waypoint
>
> *waypoints: a sequence of nodes that connect between current state $s_t$ and subgoal $sg_t$
>
> **A5-2.** Since the low-level horizon $c_l$ is determined by the temporal distance between waypoints (using equation 1 in manuscript), we can also express the $c_l$ between the $i-1$th waypont and $i$  th waypoint as $c_{l,i}$, but we omit the subscript $i$ in some statements that do not need to specify the waypoint.
>
> Reflecting on your comment, we recognized that some similar variables ( $c_l$, $c_h$, $h^{high}$ and $h^{low}$) can cause misunderstanding. Thus, we added some sentences that clearly describe the variables and omitted letters in [147-149]

---

> > ### Author Response · Authors · 2022-08-02
> > **Response to reviewer ctmE (2/2)**
> >
> > **Q6. Comparison with HiTS and HAC**
> >
> > **A6.** The following is a list of the baselines we compared: HIGL-Dense, HIGL-Sparse, HRAC-Dense, HRAC-Sparse, HIRO-Dense, HIRO-Sparse, and SAC.  Also, it should be noted that we have compared DHRL with a larger number of baselines than previous papers(HIRO,HAC,L3P) and SOTA HRL papers (HRAC, HIGL) did. HRAC adopted only two baselines  (HIRO and vanilla RL) and HIGL adopted only one baseline (HRAC).
> >
> > The reason why we didn't consider HiTS and HAC as a baseline is as follows. Since HAC is an algorithm that has been published a few years ago, it would show lower performance compared to SOTA HRL algorithms. In addition, HiTS is an algorithm specialized for dynamic environments, and It has been reported that the performance of HiTS in static environment tasks is similar to or lower than that of HAC.
> >
> > To address your concern, we also compared DHRL with additional HRL algorithms (HiTS and HAC).
> >
> > | AntMaze | 0.3M step | 0.5M step | 1.0M step | AntFourRooms | 0.3M step | 0.5M step | 1.0M step |
> > | --- | --- | --- | --- | --- | --- | --- |---|
> > | HAC | **0.195** | 0.250 | 0.350 | HAC | 0.196 | 0.589 | **0.947** |
> > | HiTS | 0.159 | 0.210 | 0.266 | HiTS | 0.000 | 0.155 | 0.701 |
> > | DHRL | 0.108 | **0.560** | **0.926** | DHRL | **0.479** | **0.689** | 0.810 |
> > - - -
> > **Q7. Concerns about section 4.3 (off-policy correction, non-stationarity of the high-level MDP, and comparison with HAC).**
> >
> > **A7-1.** By 'off-policy correction', we were referring to the method which can address the non-stationarity of the high-level MDP introduced in HIRO. A detailed explanation of off-policy correction is in L80-81 in our manuscript. As you mentioned, The main reason why DHRLs cannot be learned in an off-policy manner is the non-stationarity of the high-level MDP. What we were trying to say through the section 4.3 was that HIRO-style off-policy correction, a popular method adopted by SOTA HRL algorithms to solve this problem, is not proper for our setting.
> >
> > **A7-2.** We appreciate your helpful comment and understand your concerns about the clarity of some words in our manuscript. We rectified the whole section 4.3 in the new version. Therefore, we attach a brief summary of the revised version below. Please let us know if our responses have addressed your concerns.
> >
> > *a summary of revised version*
> >
> > This non-stationarity of the high-level MDP is caused by the presence of a graph and low-level policy. Since the graph is gradually updated, it is challenging to train a model ... Furthermore, as $c_{h}$ gets longer, predicting the similarity of the trajectory of $c_{h}$ steps with only the first step of $\pi^{low}(a_i|s_i, g_i)$ gets more difficult when we replace subgoal using HIRO-style off-policy correction. ... To facilitate the off-policy learning for longer $c_{h}$ and changing graph in DHRL, We adopt a well-known hindsight action relabelling method proposed in Hierarchical Actor-Critic (HAC) ...
> >
> > **A7-3.** Our method utilizes the optimality of the graph and low-level policy to include a graph in the learning process (Theorem 4.2 is related to this assumption), while HAC-style hindsight action relabelling method assumes the optimal low-level policy $\pi^{\mathrm{lo}*}$ only. Since it is methodically similar, so we rectified the term 'inspired' and specified the commonalities and differences. [line 218, 223]
> >
> > - - -
> > **Q8. Sensitivity analysis : threshold parameters for reaching a waypoint**
> >
> > Following your comment, we attach an additional result about the DHRL's learning sensitivity below.
> >
> > | low-level thr | 0.5 | 1.0 | 1.5 | 2.0 |
> > | --- | --- | --- | --- | --- |
> > | success rate | 0.911 | 0.854 | 0.799 | 0.695 |
> > | high-level thr | 0.5 | 1.0 | 1.5 | 2.0 |
> > | success rate | 0.911 | 0.973 | 0.938 | 0.967 |
> > - - -
> > **Q9. Suggestions in writing**
> >
> > **A9-1.** Thank you for your helpful comments to improve the clarity of our manuscript. We revised our manuscript to incorporate your comments. We clarified the vague expressions and added sentences to improve the clarity of our argument [line 110, 148, 160, 180-191, 218-225, 308-313] and [Table1, Algorithm6, Fig11 in Append A]. Please let us know if you think we missed anything. Also, regarding the two-letter variables, please pardon the delay in our revision since we think it could be more confusing if we fix it during the respond period. So we will reflect it in the final version.
> >
> > **A9-2.** We added algorithm boxes 4, 5, and 6 for each module of DHRL (FGS and gradual penalty), as well as an overview of our algorithm. We will also do our best to explain our method in detail for those who want to implement the proposed module themselves or further develop our idea.
> > - - -
> > Thank you again for your valuable and insightful review.
> >
> > Please let us know if there is anything else that we can provide for the reader's understanding. We will be happy to comply.
> > - - -
> > ###### [1] Nachum, Ofir, et al. "Why does hierarchy (sometimes) work so well in reinforcement learning?." (2019).

---

> > > ### Comment · Reviewer_ctmE · 2022-08-06
> > > **Response to authors**
> > >
> > > Thank you for your answer.
> > > My concerns have been addressed in the response.
> > > Additional clarifications have been implemented and additional experiments were conducted to solidify the results.
> > >
> > > However, I cannot see any updates to the paper or an appendix (or did I miss anything)

---

> > > > ### Author Response · Authors · 2022-08-07
> > > > **Thank you for your response (revision attached)**
> > > >
> > > > We are happy to hear that our response addressed your concerns.
> > > >
> > > > Also, we have inquired of openreview and Neurips committee about the error that the new version of the manuscript does not appear in your console.
> > > > (In the author console, the file can be downloaded normally by clicking the pdf mark.)
> > > > Until this problem is solved, following [1], we would like to attach an anonymous link to access the manuscript.
> > > >
> > > >
> > > > Anonymous URL (manuscript attached):
> > > >
> > > >
> > > > https://drive.google.com/file/d/1z23FDNj-83FX2fFSguhWJzelqA0DDuCO/view?usp=sharing
> > > >
> > > > We appreciate again for your valuable suggestions and all your efforts during the review process.
> > > >
> > > >
> > > > If you are satisfied with our response and we have addressed your concerns, we'd appreciate it if you'd please adjust your assessment to reflect your updated opinion.
> > > > If there is anything else that we can clarify, please do not hesitate to let us know. We will be happy to comply.
> > > >
> > > > ***
> > > > [1]https://nips.cc/Conferences/2022/PaperInformation/NeurIPS-FAQ/
> > > > (Can we upload a revision of our paper? Yes / Can we include an anonymous link in the author rebuttal? Yes)

---

> > > > > ### Comment · Reviewer_ctmE · 2022-08-07
> > > > > **Response**
> > > > >
> > > > > I can access the changed version through the console now.
> > > > > I would ask the authors to add the additional results etc in the final version of the paper and the appendix, if there is not enough space in the main text.

---

> > > > > > ### Author Response · Authors · 2022-08-08
> > > > > > **Thank you for your suggestions**
> > > > > >
> > > > > > Thank you for replying to our response.
> > > > > >
> > > > > > Following your suggestion, we will add the additional results in the final version. We appreciate again your efforts during the review process.

---

### Official Review · Reviewer_dQCH · 2022-07-11

**Rating:** 7
**Confidence:** 4
**Soundness:** 3 good
**Presentation:** 3 good
**Contribution:** 2 fair

**Summary:**

The method proposed in the paper suggests that supporting a decoupling between the high and low-level policy control frequency can result in better performance. Often in HRL, there are limitations in the communication between the high and low-level policy, and this communication is the source of poor performance for these models, keeping them from outperforming other methods on more complex tasks. This paper proposes the reduce over-estimation and increase generalization by introducing a method that combines graph-based methods (Like SORB) and HRL methods like HIRO to create a more versatile HRL agent with fewer learning limitations. Methods from HIRO are used to reliable data for better learning the low-level policy, and methods from SORB-like methods are used to help propose better subgoals.

--------

The score has been updated after the first responses.

-----

The score has been updated again after further discussions.

**Questions:**

Questions/Comments:

- section 4.1 discusses the particular Graph Search algorithm that is used by the method. it would be extremely helpful to the reader if the description in this area can also make more connections to Prior work to allow the reader to open understand the novelty and in particular what is different between these graph search algorithms in this section and prior work like sore
- the paragraph break online 180 is very jarring. and the next paragraph it's not clear which particular property is being discussed and mitigated.
- There does not appear to be any information related to the statistical validity of the findings in the experiments. how are the number of random seeds used in the experiments determined? what is the confidence that we have in the overall statistical tests that we are using to compare different methods?
- Figure 10 appears to only show comparisons to HIRO across a few of the scenarios. it's important to have some additional comparisons to understand how this method compares to popular prior methods.
- The claim that the paper points of a fundamental reason for HRL structures being not scalable is unclear. It will help if there is some theoretical grounding to place this contribution. Figure 3 appears to be more of a concept diagram than a fundamental reason in the optimization that outlines the limitation.

**Limitations:**

The authors note some limitations.

**Strengths And Weaknesses:**

Pros:
- The method proposed in the paper has some potential nice connection to shy the results are strong.
- HRL is challenging, and this method may be a nice step toward eliminating some of the issues with HRL methods.
- Blending high-level policies with graph-based methods provides a nice trade-off between the weakness of both.

Cons:
- The authors note that the reason their method is an improvement over prior graph-based methods is that it doesn't rely on or it can support sparse reward methods and doesn't rely on random resets. In some cases, random resets can be even more plausible and being able to reset an agent back to the same exact initial conditions given certain constraints inside of the environment. It would be very helpful if the authors can expand on this discussion in this section to help us better understand the reasoning behind the method and how it is different and an improvement over prior methods.
- One of the points argued in the paper is that being able to decouple the horizons of the mobile high-level policy should make training better. It is discussed in the paper in the introduction but it's not a fairly clear argument as to why having this disconnect is going to result in better performance. In particular, some understanding or framework that places this inside of having better generalization and/or control across the policies would likely assist the readers with appreciating the insights in this method. This would also help readers better understand when this method will be an improvement over other methods.
- To help the organization of the paper, if the earlier sections that outline additions to the method reference later parts of the experiment section, the reader can reference and understand the importance of these additions. Otherwise, as the paper is now, it's rather difficult to infer the particular contribution importance of changes over prior methods.

---

> ### Author Response · Authors · 2022-08-02
> **Response to reviewer dQCH (1/2)**
>
> Dear reviewer dQCH,
>
>
> We sincerely appreciate your constructive and insightful comments. We found them extremely helpful in improving our manuscript. We prepared our response below:
> - - -
> **Q1. Clarification on words about starting (=reset) conditions**
>
> **A1-1.** The meaning of the words 'random starting condition' and 'fixed starting condition' in this paper is as follows.
> - fixed starting condition: The agent is initialized with the distribution located at one end of the map. We implement it by adding noise to the initial robot posture and initial angular velocity.
> - random (uniform) starting condition: Initial state distribution of the agent is the uniform distribution over the feasible state space. (Thus, it needs prior knowledge about the map and obstacles)
>
> Thanks to your valuable comment, we recognized that the word 'fixed starting condition' and 'random starting condition' can cause misunderstanding. We changed each word to a 'fixed initial state distribution' and 'uniform initial state distribution'.
>
> **A1-2.** Our algorithm with uniform initial state distribution.
>
> As shown in the table below, the wider the initial distribution, the easier it is for the agent to explore the map. In other words, the 'fixed initial state distribution' condition we experimented with is a more difficult condition than the 'uniform initial state distribution'. Of course, 'fixed initial state distribution' requires less prior information about the state space. We further experimented with ours (DHRL) under various types of the reset condition. As expected, our algorithm shows faster exploration at the uniform reset point.
>
> | Level of difficulty | easy | medium | hard |
> | --- | --- | --- | --- |
> | reset point | uniform initial state distribution | few (2) reset points  | only 1 fixed initial state distribution |
> | AntMaze 0.3M  | 0.804  | 0.285 | 0.122 |
> | AntMaze 0.5M  | 0.871 | 0.882 | 0.715 |
> - - -
> **Q2. Clarification on our arguments: why does DHRL show better performance?**
>
> Thank you for your helpful suggestions on the need for better clarification of the strength of our method. We would like to provide the following in this answer: A2-1) a better explanation of our argument, A2-2) experimental results that can better reflect the strength of our concept, and A2-3) modification of the manuscript that reflects your helpful comment (we rectified some strong and vague expressions).
>
> **A2-1.** clear explanation of our argument
>
> HRL is a framework that was proposed to solve the long-horizon problem. However, in practice, previous works have demonstrated that HRL struggles to handle complex problems and it remains a challenge in the HRL community such as HRAC and HIGL (they are still trying to solve the complex problems in a data-efficient way). Although recent algorithms have improved the performance of HRLs on some long-horizon tasks, these recent algorithms have been able to solve only limited complexity problems as we introduced in section 3. This paper was motivated by these practical issues and is part of ongoing research in the HRL community to address the shortcomings of existing HRL algorithms.
>
> **A2-2.** Experimental results that can better show our concept
>
> As shown in the table below, the performance of the previous HRL is greatly affected by the high-level action interval $c_h$.
> - (In previous HRL) Particularly, if $c_h$ increases, the performance of the low-level agent decreases, and when $c_h$ decreases, the performance of the high-level agent decreases. This reflects the tradeoff in performance between the high level and the low level.
> - (In previous HRL) Due to this relationship, the longer the environment horizon, the lower the performance. (If we peg $c_h=10$, the performance plunges : 0.884 (12 * 12) -> 0.489 (24 * 24) -> 0.000 (56 * 56))
>
> In contrast, our algorithm decouples the horizon of the two levels and shows superior performance even in a large environment. Please let us know if our responses have addressed your concerns.
>
> | map size | HRAC $c_h$ = 5 | HRAC $c_h$ = 10 | HRAC $c_h$ = 30 | HRAC $c_h$ = 50 | DHRL(Ours) |
> | --- | --- | --- | --- | --- | --- |
> | 12 * 12 | 0.430 | 0.884 | 0.783 | 0.045 | 0.951 |
> | 24 * 24 | 0.180 | 0.489 | 0.574 | 0.164 | 0.911 |
> | 56 * 56 | 0.000 | 0.000 | 0.000 | 0.000 | 0.401 |
>
> **A2-3.** modification of the manuscript that reflects your helpful comment.
>
> We appreciate your helpful comment. By 'fundamental reason', we just meant 'one of the main limitations'. However, we understand your concerns about the clarity of those words in our manuscript and clarified the vague expressions such as 'fundamental' and 'extremely'. Also, we attached a new version of our manuscript that incorporates additional explanations on how DHRL can address practical issues of previous HRL methods in complex environments [Table1 in Appendix A, line 35, 146, 180, 206, 308]. Please let us know if there is anything else that we can revise.

---

> > ### Author Response · Authors · 2022-08-02
> > **Response to reviewer dQCH (2/2)**
> >
> > **Q3. Suggestion in writing: about paragraph in section 4.2 [line 180]**
> >
> > **A3.** We appreciate your helpful comments and acknowledge that there is room for improvement in the explanation of section 4.2. Following your comment, we have revised the section for a better connection between paragraphs [line 180-191].
> >
> > *a summary of the revised version*
> >
> > Unlike previous graph-guided RL methods that leverage a graph only after the actor and critic networks have been learned, we construct and utilize a graph during training. To do so, the low-level policy $Q^{lo}$ is needed to assign edge costs. However, when $\pi^{lo}$ is not yet competent in achieving some goals or encounters difficult goals, the accumulation of failure experiences in the replay buffer causes underestimation of Q-values. This leads to an overestimation of temporal distance and spoils the graph. To prevent this, we propose SQGC to construct a proper graph even when the policy is not sufficiently trained.
> > - - -
> > **Q4. Suggestion in writing : more explanation about graph search algorithm**
> >
> > To find the shortest path, we adopt Dijkstra's Algorithm, as in previous studies (SORB). There is three differences between the graph search algorithm used in the previous methods and ours.
> >
> > a) The frequency of the graph search. (In DHRL, the frequency of the graph search is equal to the frequency of high-level action.)
> >
> > b) SORB and L3P use the final goal as a goal node of the planning while ours use the action of the high level as a goal node.
> >
> > c) Since our method leverage a graph before the map is fully explored, we consider a secondary path when there is no path to the subgoal.
> >
> > Following your suggestions, we added the explanation about the difference in the graph search algorithm in section 4.1 [line 160]
> > - - -
> > **Q5. Statistical validity (the number of random seeds).**
> >
> > **A5.**  As described in the captions of Figure 6 and Table 1, success rate values are averaged over 4 random (NOT cherry-picked) seeds. We note that this number of random seeds is common practice in graph RL or HRL community (SORB and HRAC used 5 random seeds and no such information was found in the L3P and HIGL papers). Also, DHRL consistently outperformed baselines in 7 different environments.
> >
> > Nevertheless, to further address your concern, we additionally evaluated DHRL with 10 more random seeds in some environments. The following table is the result.
> >
> > |  | AntMazeSmall 0.5M | AntMazeSmall 1M | AntMaze 0.5M | AntMaze 1M |
> > | --- | --- | --- | --- | --- |
> > | DHRL success rate | 0.890 | 0.950 | 0.820 | 0.960 |
> >
> > If there are any recommendations for other validation methods that can be found in previous HRL papers, or anything we're missing, please do not hesitate to let us know as soon as possible.
> > - - -
> > **Q6. Figure 10 appears to only show comparisons to HIRO across a few of the scenarios. it's important to have some additional comparisons.**
> >
> > **A6-1.** We note that the comparison with baseline is drawn in Figure 6 and Figure 10. At first, we tried to draw every baseline in Figure 6, but because baselines overlapped with each other near the low success rate, we only left SOTA algorithms in Figure 6, and other well-established baseline algorithms were passed to the appendix (Figure 10). The following is a list of the baselines we compared: HIGL-D, HIGL-S, HRAC-D, HRAC-S, HIRO-D, HIRO-S, SAC. Also, it should be noted that we have experimented with more baselines, and in more environments than previous papers(HIRO,HAC,L3P), including SOTA HRL methods (HRAC, HIGL). HRAC adopted only two baselines(HIRO and vanilla RL) and HIGL adopted only one baseline(HRAC).
> >
> > **A6-2.** We also compared DHRL with additional HRL algorithms in some environments (HiTS and HAC).
> >
> > |AntMaze|0.3M step|0.5M step|1.0M step|AntFourRooms|0.3M step|0.5M step|1.0M step|
> > |---|---|---|---|---|---|---|---|
> > |HAC|**0.195**|0.250|0.350|HAC|0.196|0.589|**0.947**|
> > |HiTS|0.159|0.210|0.266|HiTS|0.000|0.155|0.701|
> > |DHRL|0.108|**0.560**|**0.926**|DHRL|**0.479**|**0.689**|0.810|
> > - - -
> > **Q7. Suggestion in writing**
> >
> > **A7.** Thank you for your helpful comments to improve the clarity of our manuscript. We revised our manuscript to incorporate your comments, especially (1) references in early parts, (2) changes over prior methods [line 110&Fig11 in Appendix A, 146, 160, 180, 206, 223], and (3) a detailed explanation of why DHRL outperforms previous methods [Table 1, line 308-313, algorithm 6].
> >
> >
> > If you still find it difficult to acknowledge the contribution of our algorithm, please kindly let us know which part causes the difficulty or confusion.
> > - - -
> > Thank you again for your valuable and insightful review.
> >
> > If our reply is not enough to address your concern about the contribution or the impact of our research even though it shows superior performance, please let us know if there is anything else that we can revise. We will be happy to comply.

---

> > > ### Comment · Reviewer_dQCH · 2022-08-05
> > > **Response**
> > >
> > > Thank you for this detailed response.
> > >
> > > They have clarified
> > > - The contribution better
> > > - The experimental analysis
> > > - The motivation of the algorithm and how the motivation compares to similar algorithms.
> > >
> > > It would help to explain the difference between prior work further. The method in this paper seems similar to the work in [18]. In particular, some discussion on the idea of learning a better representation between planning levels (that can be more temporally independent) and the proposed method that learns a more time independent method.
> > >
> > > [18] Soroush Nasiriany, Vitchyr Pong, Steven Lin, and Sergey Levine. Planning with goal-conditioned policies.415
> > > Advances in Neural Information Processing Systems, 32:14843–14854, 2019

---

> > > > ### Author Response · Authors · 2022-08-07
> > > > **Thank you for the response**
> > > >
> > > > We are happy to hear that our response addressed your concerns.
> > > >
> > > >
> > > > Unlike our method, LEAP [18] trains the VAE (to choose subgoals) before the RL is learned. In particular, in the locomotion task, LEAP[18] requires a dataset in which the robot is located variously within the feasible state to learn VAE (before training RL policy). In other words, LEAP also sidestepped the exploration problems, as the previous methods we mentioned in section 3.
> > > >
> > > > (In LEAP[18] paper, "The VAE is trained on a dataset in which the ant is in various valid positions of the maze, with a fixed orientation and fixed joint angles.")
> > > >
> > > >
> > > > Following your suggestion, we will discuss this in the final version. We appreciate again for your valuable suggestions and all your efforts during the review process.
> > > >
> > > > ***
> > > > [18] Soroush Nasiriany, Vitchyr Pong, Steven Lin, and Sergey Levine. Planning with goal-conditioned policies. Advances in Neural Information Processing Systems, 32:14843–14854, 2019

---

### Official Review · Reviewer_sUno · 2022-07-11

**Rating:** 7
**Confidence:** 4
**Soundness:** 3 good
**Presentation:** 3 good
**Contribution:** 2 fair

**Summary:**

In this work, the authors tackle long horizon reinforcement learning tasks by introducing a hierarchical reinforcement learning (HRL) method called DHRL. DHRL consists of 3 layers - a high-level policy that outputs a subgoal, a mid-level graph that outputs waypoints towards the subgoal, and a low-level policy that outputs actions to navigate towards the waypoints. Doing so decouples the planning horizons between the high-level and low-level policies allowing for more efficient planning of longer horizon tasks. Because the mid-level graph is not differentiable, experience relabeling is adopted to train the high- and low-level policies separately. The method is evaluated on a maze navigation and manipulator goal tracking tasks in simulation and results show significant improvement over comparison RL and HRL methods in terms of sample efficiency and final performance.

**Questions:**

- If I understand correctly, the high- and mid-level planning is similar to a probabilistic road map only instead of randomly sampling, the high-level policy provides the samples towards the goal. If this is the case, wouldn't the method be stuck at the beginning when the high-level policy is untrained? If not, then how is the graph constructed?

- what is most time consuming component of DHRL in terms of inference and how does this component scale with the dimensionality of its input?

- I would suggest avoiding overly strong but vague claims such as "fundamental reason", "extremely complex and long tasks"


**Limitations:**

Having a graph middle layer inherits some of the limitations of graph-based planning methods. This includes the inefficiency caused by (1) potentially high dimensional state space (mentioned by the authors in Section 6); (2) closely situated obstacles or corridors as in a cluttered environment. Predicates can also be somewhat limited in terms of representing safety in the observation space.

**Strengths And Weaknesses:**

Strengths
- The method is conceptually sound and the improvements on the chosen tasks are significant. I also liked the details and practical considerations the authors included in their discussions.

Weaknesses
- Navigation task and goal tasks are conceptually simple. Long horizon tasks should be accompanied by increased complexity and practicality in the real world. The authors are encouraged to try DHRL on kitchen tasks (https://github.com/rmrafailov/kitchen ) and more complex manipulation tasks (GitHub - stepjam/RLBench: A large-scale benchmark and learning environment. )

---

> ### Author Response · Authors · 2022-08-02
> **Response to reviewer sUno (1/2)**
>
> Dear reviewer sUno.
>
>
> We sincerely appreciate your constructive and insightful comments. We found them extremely helpful in improving our manuscript. We prepared our response below:
> - - -
> #### **Q1. Navigation tasks and goal tasks are conceptually simple.**
>
> **A1.** We appreciate your comment. However, it should be noted that we have tested DHRL on **almost every task** used in the SOTA paper [1]. Furthermore, we proposed new challenging tasks which SOTA HRL algorithms have failed such as UR3Obstacle, AntMazeComplex, and Bottleneck. Please consider that DHRL consistently shows superior performance in (a)environments widely used in the HRL community (typical HRL tasks) as well as (b)newly proposed challenging tasks (long & sparse tasks).
>
> |  | DHRL(Ours) | HIGL-D | HIGL-S | HRAC-D | HRAC-S | HIRO-D | HIRO-S | SAC |
> | --- | --- | --- | --- | --- | --- | --- | --- | --- |
> | PointMaze | **0.998** | 0.883 | 0.918 | 0.335 | **0.937** | 0.936 | 0.768 | 0.003 |
> | AntMazeSmall | **0.951** | 0.837 | 0.522 | 0.834 | **0.884** | 0.241 | 0.601 | 0.000 |
> | AntMaze | **0.911** | **0.781** | 0.603 | 0.687 | 0.489 | 0.007 | 0.008 | 0.000 |
> | Bottleneck | **0.387** | 0.000 | 0.000 | 0.000 | 0.000 | 0.000 | 0.000 | 0.000 |
> | Complex | **0.401** | 0.000 | 0.000 | 0.000 | 0.000 | 0.000 | 0.000 | 0.000 |
> | Reacher-3D | **0.906** | - | 0.471 | - | 0.441 | - | **0.671** | 0.472 |
> | UR3Obstacle | **0.698** | **0.111** | 0.005 | 0.005 | 0.000 | 0.025 | 0.019 | 0.000 |
> - - -
> #### **Q2. Wouldn't the method be stuck at the beginning when the high-level policy is untrained? & How is the graph constructed?**
>
> Of course, as most graph-RL methods do, DHRL inherits some traits of PRM. However, it has superior performance to other graph-RLs and is not initially stuck due to the following reasons.
>
> **A2-1.** DHRL does not get stuck at the beginning.
>
> high-level agent: If the high-level agent delivers a subgoal but the low level could not achieve it, the high level receives a penalty (line14 in algorithm 1), so the high-level policy can learn to give an achievable goal distribution. Of course, some amount of initial training step is required for the high-level agent to learn this, and this is a common characteristic of HRL. Please note that the number of training steps (x-axis) in the graphs and tables in the manuscript includes the steps corresponding to these initial training steps (to evaluate our algorithm fairly).
>
> Low-level agent: Even if the high level gives a goal that is not feasible or achievable, the low-level policy can learn how to achieve the nearby goals that the low level has achieved unintentionally through hindsight goal relabeling.
>
> **A2-2.** How to construct an initial graph?
>
> Since the graph requires a learned Q-function, the first few steps (what we call the 'initial training step' in this answer) run like normal HRLs without graphs or waypoints. After that, a graph is formed, from which the waypoint is generated along the FPS algorithm (algorithm 2 in appendix A) and Dijkstra's algorithm. We can choose the number of 'initial training steps', but it does not greatly affect the performance, so we maintained the same value (75 episodes) across various environments. We added a new algorithm box to represent this process (See lines 11-26 of Algorithm 6).
> - - -
> #### **Q3. What is the most time-consuming component of DHRL in terms of inference?**
>
> **A3.** The following table shows the inference speed of each module in DHRL. (the experiment was processed using an NVIDIA RTX A5000 and AMD Ryzen 2950x in an AntMaze environment.)
>
> |  | frequency | time(ms) |
> | --- | --- | --- |
> | Add $s_t$ and $sg_t$ to graph layer | 10 per 1 episode (600steps) | 4ms |
> | Dijkstra's algorithm (300nodes) | 10 per 1 episode (600steps) | 0.2ms |
> | query action to high-level policy | 1 per 60 steps | 0.5ms |
> | query action to low-level policy | 1 per 1 step | 0.5ms |
>
> This experiment indicates that low-level policy consumes the longest time per step. We note that other RL algorithms also require these neural network inference time every step.
>
> - - -
> ###### [1] Junsu Kim, Younggyo Seo, and Jinwoo Shin. Landmark-guided subgoal generation in hierarchical reinforcement learning. Advances in Neural Information Processing Systems, 34, 2021.

---

> > ### Author Response · Authors · 2022-08-02
> > **Response to reviewer sUno (2/2)**
> >
> > #### **Q4.** **Does the graph-based planning method include the inefficiency caused by closely situated obstacles or corridors as in a cluttered environment?**
> >
> > **A4.** We acknowledge the generally known limitations of the graph. However, practically, we could not find additional performance degradation of DHRL caused by using graphs compared to conventional RL in cluttered environments.
> >
> > We cite the AntMazeBottleneck environment in our manuscript as an example of a cluttered environment. The results in the manuscript show that the HRL and single-layer RL baselines do not succeed at all, and only DHRL succeeded in this task. Through these experiments, we conclude that a) the performance bottleneck of training RL in those environments could be exploration (or data collection), not path planning, and b) DHRL solves those environments better than the previous RL methods in the cluttered environment because the graph has improved the exploration performance.
> > - - -
> > #### **Q5. About "somewhat limited in terms of representing safety in the observation space"**
> >
> > **A5.** DHRL does not specify or represent safety in the observation space. Since we use only a critic network to construct a graph, representations that DHRL learns are not different from the general HRL methods. Thus, there are no additional limitations related to 'representing safety' that DHRL has. If we misunderstood your comment, please let us know.
> > - - -
> > #### **Q6. Suggestion in writing**
> >
> > **A6.** We appreciate your helpful comment. By 'fundamental reason', we just meant 'one of the main limitations'. However, we understand your concerns about the clarity of those words in our manuscript and clarified the vague expressions such as 'fundamental' and 'extremely'. Please let us know if there is anything else that we can revise.
> > - - -
> > #### **A7. Experimental results that can better show our concept**
> >
> > We would like to report an additional result that can better show our concept. As shown in the table below, the performance of the previous HRL is greatly affected by the high-level action interval $c_h$.
> >
> > - (In previous HRL) Particularly, if $c_h$ increases, the performance of the low-level agent decreases, and when $c_h$ decreases, the performance of the high-level agent decreases. This reflects the tradeoff in performance between the high level and the low level.
> > - (In previous HRL) Due to this relationship, the longer the environment horizon, the lower the performance. (If we peg $c_h=10$, the performance plunges : 0.884 (12 * 12) -> 0.489 (24 * 24) -> 0.000 (56 * 56))
> >
> > In contrast, our algorithm decouples the horizon of the two levels and shows superior performance even in a large environment.
> >
> > |  map size | HRAC $c_h$ = 5 | HRAC $c_h$ = 10 | HRAC $c_h$ = 30 | HRAC $c_h$ = 50 | DHRL(Ours) |
> > | --- | --- | --- | --- | --- | --- |
> > | 12 * 12 | 0.430 | 0.884 | 0.783 | 0.045 | 0.951 |
> > | 24 * 24 | 0.180 | 0.489 | 0.574 | 0.164 | 0.911 |
> > | 56 * 56 | 0.000 | 0.000 | 0.000 | 0.000 | 0.401 |
> >
> > - - -
> > Thank you again for your valuable and insightful review.
> >
> > Please let us know if our responses have addressed your concerns. If anything needs further clarification, please do not hesitate to let us know as soon as possible.

---

> > > ### Author Response · Authors · 2022-08-07
> > > **Revision attached**
> > >
> > > We hope you are doing well.
> > >
> > > Just in case you cannot access the new version of the manuscript (since some reviewers say they can't download the revision in reviewer console), we would like to attach an anonymous link to access the manuscript (until this problem is solved; we reported it to openreview and Neurips team). We appreciate again for your valuable suggestions and all your efforts during the review process.
> > >
> > >
> > > Anonymous URL (manuscript attached):
> > >
> > >
> > > https://drive.google.com/file/d/1z23FDNj-83FX2fFSguhWJzelqA0DDuCO/view?usp=sharing
> > >
> > >
> > > ***
> > > [1]https://nips.cc/Conferences/2022/PaperInformation/NeurIPS-FAQ/
> > > (Can we upload a revision of our paper? Yes / Can we include an anonymous link in the author rebuttal? Yes)

---

> > > > ### Comment · Reviewer_sUno · 2022-08-07
> > > > **Response**
> > > >
> > > > I thank the authors for addressing my questions. Upgrade score to 7.

---

> > > > > ### Author Response · Authors · 2022-08-08
> > > > > **Thank you for the response**
> > > > >
> > > > > Thank you for replying to our response. We are happy to hear that our response addressed your questions.
> > > > > We appreciate again your efforts during the review process.

---

### Official Review · Reviewer_B21k · 2022-07-12

**Rating:** 7
**Confidence:** 3
**Soundness:** 3 good
**Presentation:** 3 good
**Contribution:** 3 good

**Summary:**

Paper proposes a new method for hierarchical RL which decouples the horizon of the high level policy (outputs subgoals) from the horizon of the low level policy (outputs actions given target waypoint), by adding an additional planning phase between the high level and low level policies which searches over a graph for a sequence of waypoints which reaches the subgoal. The experiments show that this decoupling of horizons allows RL algorithms to better solve long-horizon tasks, and that the proposed algorithm outperforms prior HRL algorithms.

**Questions:**

N/A

**Limitations:**

Authors noted that constructing the graph used for planning may be difficult for environments with high-dimensional state spaces, and proposes to instead learn the graph over a latent space.

No negative societal impact that I am aware of.

**Strengths And Weaknesses:**

Strengths
- Nicely written paper which clearly explains the difficulties of using prior HRL algorithms (horizon of low-level and high-level policies are coupled, making it difficult to scale to very long horizon tasks), and proposes method of address this weakness by decoupling the horizon of the high-level and low-level policies
- Experiments clearly show improvements over previous HRL algorithms on a variety of tasks
- Thorough ablation studies that illustrate the importance of different components in the algorithm

Weaknesses
- Description of method is a bit hard to digest because the algorithm involves many parts, and each part involves many procedures. Maybe adding some algorithm boxes which summarize each section, and also an overarching algorithms box which connects all the subparts will aid in understanding.

---

> ### Author Response · Authors · 2022-08-02
> **Response to reviewer B21k**
>
> Dear reviewer B21k,
>
>
> We sincerely appreciate your constructive and insightful comments. We found them extremely helpful in improving our manuscript. We prepared our response below:
> - - -
> #### **Q1. Adding algorithm boxes**
>
> **A1.** Thank you for your helpful suggestions on the need for a summary of the overall algorithm. We added Algorithms 4, 5, and 6 for the gradual penalty module, frontier-based goal-shifting module, and overview of our algorithm, respectively (Appendix A in the new version). We will also do our best to explain our method in detail for those who want to implement the proposed module themselves or further develop our idea. Please let us know if our responses have addressed your concerns.
> - - -
> #### **Q2. Suggestion in writing**
>
> **A2.** We understand your concerns that the explanation of each part is rather difficult. Following your comments, we revised some parts and attached a new version of our manuscript. Please let us know if there is anything else that we can revise.
>
> - Rectified some vague words
> - Clarified section 4, 4.1, 4.2, 4.3, 4.4
>     * detailed explanations about $c_l$ and $c_h$ [L142-149]
>     * specified the commonalities and differences of the graph level over previous algorithms [L160-162]
>     * detailed explanations about why we need separated Q networks [L180-191]
>     * detailed explanations about how to train high-level policy in an off-policy manner [L212-225]
>     * added reference (newly added algorithm boxes) [L254, 263]
> - - -
> #### **A3. Experimental results that can better show our concept**
>
> We would like to report an additional result that can better show our concept. As shown in the table below, the performance of the previous HRL is greatly affected by the high-level action interval $c_h$.
>
> - (In previous HRL) Particularly, if $c_h$ increases, the performance of the low-level agent decreases, and when $c_h$ decreases, the performance of the high-level agent decreases. This reflects the tradeoff in performance between the high level and the low level.
> - (In previous HRL) Due to this relationship, the longer the environment horizon, the lower the performance. (If we peg $c_h=10$, the performance plunges : 0.884 (12 * 12) -> 0.489 (24 * 24) -> 0.000 (56 * 56))
>
> In contrast, our algorithm decouples the horizon of the two levels and shows superior performance even in a large environment.
>
> |  map size | HRAC $c_h$ = 5 | HRAC $c_h$ = 10 | HRAC $c_h$ = 30 | HRAC $c_h$ = 50 | DHRL(Ours) |
> | --- | --- | --- | --- | --- | --- |
> | 12 * 12 | 0.430 | 0.884 | 0.783 | 0.045 | 0.951 |
> | 24 * 24 | 0.180 | 0.489 | 0.574 | 0.164 | 0.911 |
> | 56 * 56 | 0.000 | 0.000 | 0.000 | 0.000 | 0.401 |
>
> - - -
> Thank you again for your valuable and insightful review.
>
> Please let us know if there is anything else that we can provide for the reader's understanding. We will be happy to comply.

---

> > ### Comment · Reviewer_B21k · 2022-08-08
> > **Response to authors**
> >
> > Thanks for the response. I will be keeping my rating at 7.

---

> > > ### Author Response · Authors · 2022-08-09
> > > **Thank you for the response**
> > >
> > > Thank you for replying to our response. We appreciate again your efforts during the review process.

---

### Meta-Review · Area_Chair_TUrS · 2022-08-25

**Recommendation:** Accept
**Confidence:** Certain

**Metareview:**

The reviewers appreciated the authors' response with additional experiments and clarifications. Given the feedback from the reviewers and the discussion, I would like to recommend this paper for acceptance and congratulate the authors on a strong submission. I encourage the authors to address the reviewers' comments for the final version of the paper.



**Award:**

No

---

### Decision · Program_Chairs · 2022-09-14

Accept